# A toxin-deformation dependent inhibition mechanism in the T7SS toxin-antitoxin system of Gram-positive bacteria

Yongjin Wang[1,7], Yang Zhou[1,7], Chaowei Shi[2,7], Jiacong Liu[1], Guohua Lv[3], Huisi Huang[1], Shengrong Li[1], Liping Duan[1], Xinyi Zheng[1], Yue Liu[1], Haibo Zhou [1], Yonghua Wang[4], Zhengqiu Li[1], Ke Ding [1] ✉, Pinghua Sun [1] ✉, Yun Huang [5] ✉, Xiaoyun Lu[1] ✉ & Zhi-Min Zhang [1,6] ✉

Toxin EsaD secreted by some *S. aureus* strains through the type VII secretion system (T7SS) specifically kills those strains lacking the antitoxin EsaG. Here we report the structures of EsaG, the nuclease domain of EsaD and their complex, which together reveal an inhibition mechanism that relies on significant conformational change of the toxin. To inhibit EsaD, EsaG breaks the nuclease domain of EsaD protein into two independent fragments that, in turn, sandwich EsaG. The originally well-folded ββα-metal finger connecting the two fragments is stretched to become a disordered loop, leading to disruption of the catalytic site of EsaD and loss of nuclease activity. This mechanism is distinct from that of the other Type II toxin-antitoxin systems, which utilize an intrinsically disordered region on the antitoxins to cover the active site of the toxins. This study paves the way for developing therapeutic approaches targeting this antagonism.

Persistently colonizing the nares of approximately 20% of humans, *S. aureus* is an opportunistic bacterial pathogen that causes bacteremia and infective endocarditis as well as osteoarticular, skin and soft tissue, pleuropulmonary, and device-related infections[1]. Due to the abusive usage of antibiotics, development of drug-resistant bacteria, such as *S. aureus* strains with methicillin-resistant or reduced susceptibility to vancomycin, has imposed a major challenge in the management of bacterial infections[2–4]. It is therefore highly desirable to develop novel antibiotics targeting new signaling pathways in *S. aureus*.

During infection, *S. aureus* secrets a wide array of effector proteins into the extracellular milieu. The primary function of these proteins is to involve in a process termed effector-targeted pathway (ETP) which manipulates key cellular processes such as RNA silencing, vesicle

trafficking, transcription, cell signaling and innate immunity of host cells[5,6], leading to perturbation of host cellular responses and establishment of an environmental niche for the bacteria in which to thrive[7]. Produced inside the bacteria, effector proteins rely on an array of secretion systems to travel into host cells or surrounding environment. In Gram-negative bacteria, most proteins are exported across the inner and outer membranes in a single step via the type I, type III, type IV or type VI secretion pathways. While the others are first translocated into the periplasmic space via the Sec or two-arginine (Tat) pathways before being exported across the outer membrane via the type II or type V secretion system[8,9]. In contrast, few specialized secretion system has been found in Gram-positive bacteria, in which proteins are commonly secreted across the single membrane by the universal Sec

[1]International Cooperative Laboratory of Traditional Chinese Medicine Modernization and Innovative Drug Development of Chinese Ministry of Education (MOE), College of Pharmacy, Jinan University, Guangzhou 510632, China. [2]Hefei National Laboratory for Physical Sciences at the Microscale, University of Science and Technology of China, Hefei 230026, China. [3]Division of Histology & Embryology, Medical College, Jinan University, Guangzhou 510632, China. [4]School of Food Science and Engineering, South China University of Technology, Guangzhou 510640, China. [5]Department of Physiology & Biophysics, Weill Cornell Medicine, New York, NY 10065, USA. [6]Guangdong Youmei Institute of Intelligent Bio-manufacturing, Foshan, Guangdong 528200, China. [7]These authors contributed equally: Yongjin Wang, Yang Zhou, Chaowei Shi. ✉e-mail: dingke@jnu.edu.cn; pinghuasunny@163.com; yuh2010@med.cornell.edu; luxy2016@jnu.edu.cn; 1363210756@163.com

or Tat pathways[8,10]. Recently, the type VII secretion system (T7SS) was identified in *mycobacterium tuberculosis*[11], and later in several Gram-positive bacteria, including *S. aureus*, *Ballus anthracis*, *Streptococcus pneumonia*, *Corynebacterium diphtheria*, and *Streptococcus agalactiae*[12,13].

Proteins secreted by T7SS have been shown to be required for the virulence of Gram-positive bacteria[14,15]. For example, both EsxA and EsxB are substrates of T7SS in *S. aureus*. EsxA interferes with *S. aureus*-induced apoptotic pathways and, together with EsxB, facilitates the release of *S. aureus* from the host epithelial cells[16]. Additionally, *S. aureus* bearing certain EsxA or EsxB mutants are defective in the

pathogenesis of murine abscesses[17]. Previous studies discovered yet another role of T7SS in microbial competition[18]. EsaD is a toxin secreted in a T7SS machinery-dependent manner by *S. aureus*. The toxic activity of EsaD is neutralized in cytoplasm by an antitoxin, EsaG, which binds to the C-terminal nuclease domain of EsaD[19,20] (Fig. 1a). During the secretion process, EsaG is stripped off from EsaD and left behind in the cytoplasm, while EsaD is transported to the outside to acts on other bacteria, killing the EsaG-deficient *S. aureus* strains[19]. These studies uncovered the antibacterial activity of T7SS substrates, implying the potential usage of EsaD in clinical therapeutics against T7SS containing pathogens, including the antibiotic-resistant *S. aureus*.

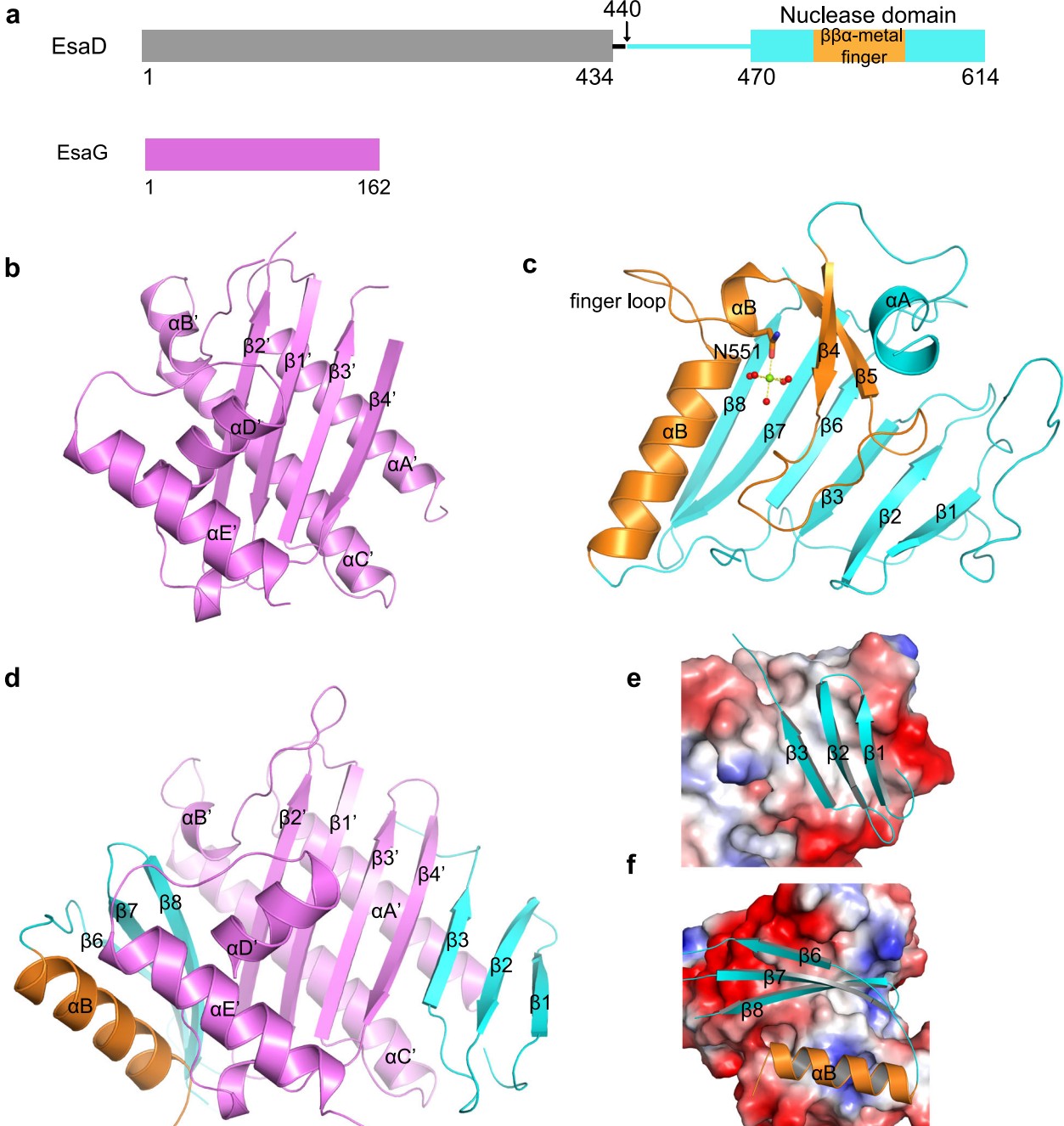

**Fig. 1 | Structure of the nuclease domain of EsaD and EsaDc-EsaG complex.** **a** Domain architecture of EsaD and EsaG, with individual domains colored differently. Similar color schemes are used in the other figures unless otherwise indicated. **b** Crystal structure of EsaG. The α-helices and β-strands are labeled from A′ to E′ and from 1′ to 4′, respectively. **c** Crystal structure of the nuclease domain of EsaD. The α-helices and β-strands are labeled from A to B and from 1 to 8, respectively. The magnesium ion (green sphere) is coordinated to an oxygen atom of N551 and five waters (red spheres). **d** Crystal structure of EsaDc-EsaG complex in one asymmetric unit. The NF binding site (**e**) and CF binding site (**f**) on EsaG are shown in the expanded view.

However, the molecular mechanism underlying EsaD-EsaG interaction remains elusive. EsaD-EsaG seems to belong to the Type II toxin-antitoxin modules, in which the antitoxin inhibits the biochemical activity of the toxin via complex formation with the toxin[21–24]. A typical feature in the toxin-neutralizing domain of the antitoxins is an intrinsic disordered region (IDR) that folds upon the toxin binding and wraps around the toxin to cover its substrate interaction site.

Herein, through the crystal structures of EsaG, the C-terminal nuclease domain of EsaD and their complex, we reveal a distinct inhibition strategy in *S. aureus*. This inhibitory mechanism depends on the deformation of the toxin, where EsaG splits EsaD into two separated fragments, resulting in the catalytic ββα-metal finger motif connecting the two fragments transformed from an originally well-folded structure into a disordered loop. This finding will improve our understanding of the mechanism of microbial interactions and may inform the design of strategies to manipulate Gram-positive bacteria for medical purposes.

## Results

### Crystal structure of EsaG implies an inhibitory mechanism independent of IDR

To investigate whether EsaG contains an IDR domain as most of the other type II antitoxins, we determined the crystal structure of full-length EsaG at a resolution of 2.30 Å (Fig. 1b and Supplementary Table 1). There are two EsaG molecules in the asymmetric unit, which interact with each other in a 2-fold symmetry. To investigate the oligomer state of EsaG in solution, we performed gel-filtration experiments, chemical cross-linking assay and density gradient centrifugation analysis. Even though gel-filtration experiments indicated a dimer state (Supplementary Fig. 1a), chemical cross-linking of EsaG using an amine-specific crosslinking reagent EGS (ethylene glycol bis(succinimidyl succinate)) yielded no cross-linked product (Supplementary Fig. 1b). Density gradient centrifugation analysis also supported a monomer form (Supplementary Fig. 1c). The structure of EsaG contains a principle four-stranded β-sheet sandwiched by αA′, αB′ and αC′ from one side and αD′ and αE′ from the other side (Fig. 1b). Structural comparison of EsaG with other protein structures using the Dali server[25] indicated a significant homology with the structure of BH3703 (PDB code 3IOT), with a Z-score of 18.9 and a root-mean-square deviation (RMSD) of 1.9 Å (Supplementary Fig. 2). BH3703 is also a putative antitoxin produced by the Gram-positive bacterium *Alkalihalobacillus okuhidensis*, but its biological function remains unknown. We noticed that, except for three residues (G50–S51 on loop$_{β2'-αB'}$ and E162 on the C-terminus), electron densities were observed for all the other residues of EsaG. Similarly, BH3703 is also well structured with only three residues in the C-terminus missing. This suggests the absence of IDRs in EsaG-like antitoxins and, therefore, a distinct toxin inhibition mechanism in Gram-positive bacteria.

### Significant conformational changes on EsaD upon EsaG binding

To investigate the mechanism by which EsaG inhibits EsaD, we next solved the crystal structures of EsaD and the EsaD-EsaG complex. For EsaD, we focused on its nuclease domain, because it has been reported that the interaction of EsaG with the nuclease domain is sufficient to neutralize the toxicity of EsaD[19]. Due to cytotoxicity, the EsaD proteins used for crystallization were over-expressed in *E. coli* in an inactive state by mutating the catalytic H528 residue to Alanine (H528A). The crystal structures were determined at resolutions of 2.30 Å for the nuclease domain of EsaD (residues 440–614) and 2.60 Å for the EsaG-EsaD (residues 450–614, hereafter referred as EsaDc) complex, respectively (Fig. 1c, d and Supplementary Table 1).

The nuclease domain of EsaD presents a typical fold of ββα-metal nucleases (also known as His-Me finger nucleases)[26], even though they share low sequence similarity (<20%). The structure contains a central 6-stranded β-sheet that is largely open to solvent on one side and

flanked by one short α-helix (αA) and the well-known ββα-metal finger motif on the other side (Fig. 1c). The ββα-metal finger (Residues 501–573), playing a critical role in nicking DNA, creates a compact catalytic center that is composed of two antiparallel β-strands (β4 and β5) connected by a long loop and linked to a C-terminal α-helix (αB). As observed in some other His-Me finger nucleases, αB is interrupted by a protruding 'finger loop' of unknown function[27]. Buried within the catalytic site, a hydrated Mg$^{2+}$ ion is directly coordinated by a highly conserved Asn residue (N551) located in αB[28–30]. In addition, a long N-terminal loop (Residues 455–468) wraps around the central β-sheet from the β1 side and extends to the "back" face.

Except for subtle movements in some loops, no obvious conformational change happens on EsaG upon EsaDc binding (Supplementary Fig. 3), further supporting the absence of IDR in the inhibition process. However, the toxin, EsaD, undergoes significant conformational changes during the complex formation. The central 6-stranded β-sheet of EsaDc is split into two independent fragments (Fig. 1d–f): the N-terminal fragment (NF) consists of β1-3, which merges with the central β-sheet of EsaG through β4′ of EsaG and β3 of EsaDc; In the C-terminal fragment (CF), αB from the ββα-metal finger and β6-8 interact with EsaG from the other side of the central β-sheet of EsaG. We could not trace the Mg$^{2+}$ ion in the catalytic center, the long loop flanking the N-terminus of NF (Residues 450–468) and part of the ββα-metal finger motif, including αA to the finger loop (Residues 504–556). Noticeably, the N-terminus and C-terminus of the missing ββα-metal finger motif locate on two opposite sides of EsaG, with a linear distance of about 44 Å (Supplementary Fig. 4). These observations indicated that the separation of NF and CF of EsaDc by EsaG may stretch the originally folded ββα-metal finger motif to make it become a disordered loop and stride across such a long distance, thus abolishing the DNase activity of EsaDc.

### The interactions between EsaDc and EsaG

All the secondary structure elements of EsaDc are involved in the direct association with EsaG, mainly through hydrogen bonds, salt bridges and hydrophobic packing interactions (Fig. 2a–c). In the NF binding surface, residue Y476$^{EsaDc}$ on β1-strand forms a hydrogen bond with the side chain of residue E4$^{EsaG}$. The carbonyl oxygen of E4$^{EsaG}$ engages two water-mediated hydrogen bonds with Y484$^{EsaDc}$ and H482$^{EsaDc}$ alone the β2-strand, respectively. Additionally, Y484$^{EsaDc}$ donates a second hydrogen bond to Y11$^{EsaG}$. Moreover, residues on the β3-strand, including V495$^{EsaDc}$ and V497$^{EsaDc}$, form a hydrophobic core with the side chains of Y11, L110, V112 and F114 of EsaG (Fig. 2a).

The CF binding is contributed collectively by αB and β6-8 of EsaDc, burying a solvent-accessible surface area of 3825 Å$^2$, which is much larger than that of the 2461 Å$^2$ in the NF binding site. Briefly, αB makes a hydrogen bond network to αD′ and αE′ of EsaG (Fig. 2b). The acidic side chain of E566$^{EsaDc}$ forms a bidentate interaction with the backbone amino groups of I127$^{EsaG}$ and G128$^{EsaG}$ on αD′ at the same time. αD′ forms a third hydrogen bond with αB through the side chains of R129$^{EsaG}$ and N562$^{EsaDc}$. In addition to these interactions, αB is further recognized by Y146$^{EsaG}$ on αE′ and D36$^{EsaG}$ on β1′. On the other side of the central β-sheet of EsaG, the β6-8 sheet is accommodated mainly by αB′ and αC′ of EsaG (Fig. 2c). β6 donates one hydrogen bond through E583$^{EsaDc}$ to Q69$^{EsaG}$ on αC′. E601 on β7 forms a salt bridge with K62$^{EsaG}$. On β8, two Arg residues (R606 and R610) form salt bridges with two Asp residues on αC′ (D72 and D80). Furthermore, the backbone carbonyl group of R606 and the side chain of N607 form hydrogen bond with T58$^{EsaG}$ and Y56$^{EsaG}$ from αB′, respectively.

### Validation of EsaDc-EsaG interactions

To test whether EsaDc interacts with EsaG through two separated fragments in solution as observed in the structure, we first performed bio-layer interferometry (BLI) experiments (Supplementary Fig. 5). Substitution of Y484$^{EsaDc}$ from the NF-EsaG interface with an alanine

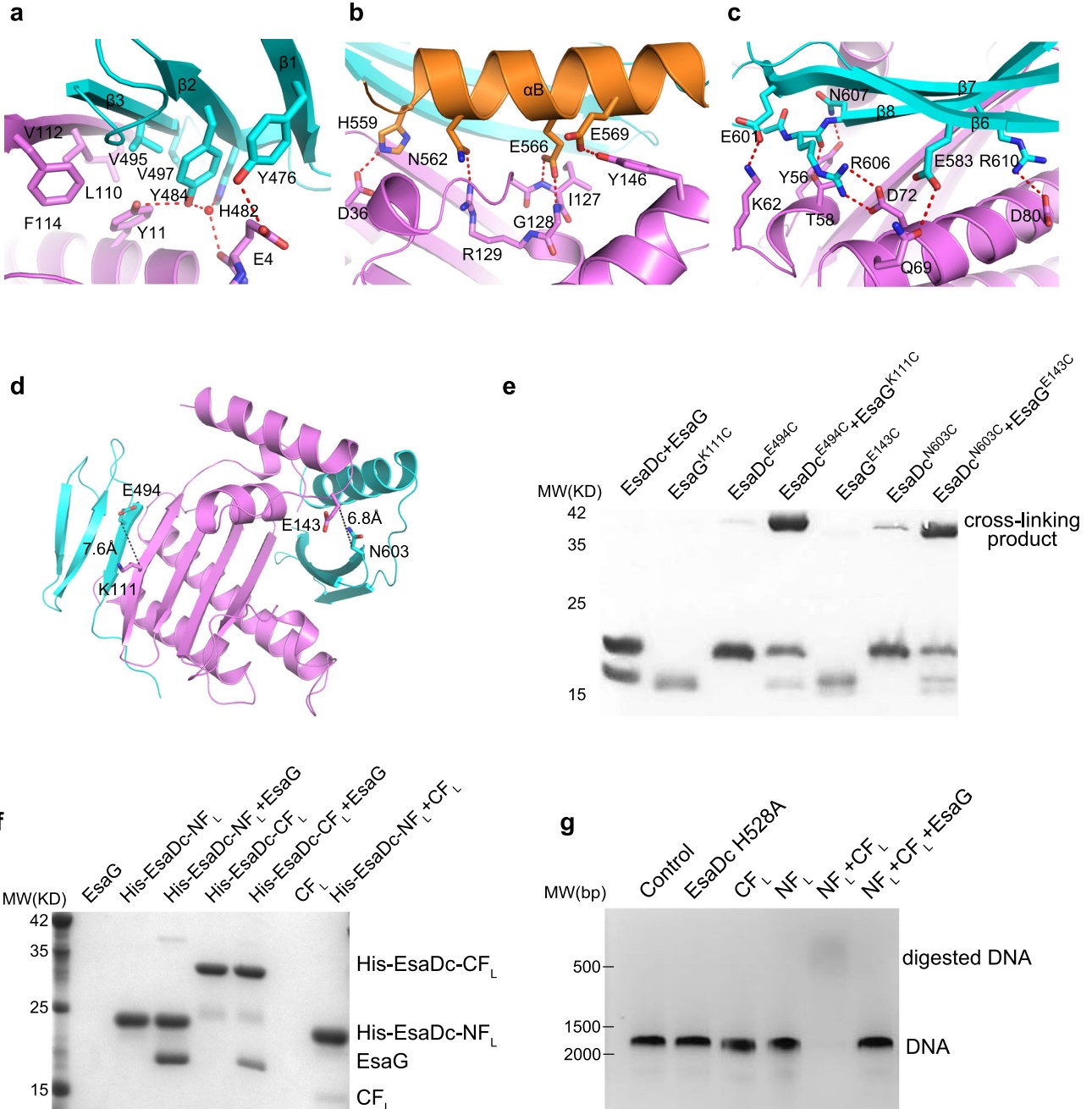

**Fig. 2 | The interactions between EsaDc and EsaG. a–c** Close-up view of the NF–EsaG (**a**), αB–EsaG (**b**) and the β-sheet (in CF)–EsaG (**c**) interactions. The proteins are colored in the same scheme as shown in Fig. 1a. The side chains of the interface residues are labeled and shown as sticks. The hydrogen bonds are depicted as red dashed lines, and the water molecules are shown as red spheres. **d** Pairs selected for cross-linking assay. EsaG$^{K111}$ and EsaDc$^{E494}$ are selected on the NF-EsaG interface. The closest distance between Cβ of EsaG$^{K111}$ and EsaDc$^{E494}$ is 7.6 Å; EsaG$^{E143}$ and EsaDc$^{N603}$ are selected close to the CF-EsaG interface. The closest distance between the atoms on the side chains of EsaG$^{E143}$ and EsaDc$^{N603}$ is 6.8 Å. **e** SDS-

PAGE analysis of EsaG and EsaDc variants incubated with the chemical probe. The cross-linked product has been labeled. **f** His-tag pull down assay. All the proteins were purified from *E. Coli*. A His-SUMO tag was co-expressed on the N-terminus of EsaDc-NF$_L$ (His-EsaDc-NF$_L$) and EsaDc-NF$_L$ (His- EsaDc-CF$_L$) to chelate with the nickel beads. Proteins are indicated by Coomassie blue staining. **g** in vitro nuclease activity of NT-CT complex. The gel was stained with ethidium bromide and visualized under UV light. The experiments were performed three times with similar results.

reduced the binding of EsaDc and EsaG to a $K_d$ of 1.67 μM, about three times lower than that of the wild-type ($K_d = 0.46$ μM). On the CF-EsaG interface, mutant N562F slightly decreased the binding affinity to a $K_d$ of 0.86 μM.

Cross-linking assays were next carried out to further confirm the two interfaces. We selected pairs of residues surrounding the interfaces for which Cβ–Cβ distances were less than 15 Å. These residues should be accessible to the solvent and not located in flexible regions.

Based on these criteria, we identified the pairs EsaG$^{K111}$- EsaDc$^{E494}$ around the NF-EsaG interface and EsaG$^{E143}$- EsaDc$^{N603}$ around the CF-EsaG interface (Fig. 2d). We then introduced cysteine residues at these positions and incubated the resulting complexes with a chemical cross-linking probe. The probe contains two electrophilic iodoacetamide (IA) groups that are separated by a 4-carbon spacer arm (Supplementary Fig. 6). IA probe could selectively target and irreversibly cross-link cysteine in native proteins[31]. As shown in Fig. 2e, EsaG or

EsaDc variants alone generated undetectable to week cross-linking signals, but the signals were drastically increased in both the pairs EsaG$^{K111C}$-EsaDc$^{E494C}$ and EsaG$^{E143C}$-EsaDc$^{N603C}$. The cross-linked proteins were recovered from SDS-PAGE, digested by trypsin and analyzed by liquid chromatography-MS (LC-MS) experiments. We successfully identified the expected cross-linked peptides in both samples as dominant cross-linked peptides (Supplementary Fig. 7 and Supplementary Data 1), supporting that the residues of selected pairs are in proximity as observed in the structure.

We also performed a His-tag pull-down assay to investigate the interaction of EsaG with each of the two fragments (Fig. 2f). We generated the two fragments of EsaDc in longer versions: the NF plus the missing loop preceding it (NF$_L$, residues 450–503), and the CF plus the disordered ββα-metal finger motif (CF$_L$, residues 504–614). The two fragments were overexpressed with an N-terminal attached His-SUMO tag and referred to as His-EsaDc-NF$_L$ and His-EsaDc-CF$_L$, respectively. As shown in Fig. 2f, both His-EsaDc-NF$_L$ and His-EsaDc-CF$_L$ can pull down EsaG efficiently. Consistently, gel-filtration analysis showed that NF$_L$ and CF$_L$ could form stable complexes and eluted with EsaG, respectively (Supplementary Fig. 8). Together, these results validate the insertion of EsaG into the central β-sheet of EsaDc and that the two fragments of EsaDc generated during the insertion could interact with EsaG independently.

### NF$_L$ and CF$_L$ reassemble into an active protein in the absence of EsaG

During the secretion process, EsaG is left behind in the cell, while EsaD is transported to surrounding environment. We thus speculated that the NF$_L$ and CF$_L$ could reassemble into an intact structure of EsaDc when EsaG is removed from the complex. In the His-tag pull-down assay, we observed that CF$_L$ associates with the His-tagged NF$_L$ (Fig. 2f), indicative of formation of intact EsaDc protein. Further evidence for this idea comes from the NMR experiments. We collected two-dimensional $^1$H,$^{15}$N-HSQC spectra for $^{15}$N-labeled NF$_L$ in the presence of unlabeled CF$_L$, $^{15}$N-labeled CF$_L$ in the presence of unlabeled NF$_L$ and $^{15}$N-labeled EsaDc, respectively (Supplementary Fig. 9). Most signals of the $^{15}$N-labeled NF$_L$ and CF$_L$ could overlap with the corresponding signals of EsaDc, supporting that the NF$_L$-CF$_L$ complex and EsaDc share the same structure. To test whether the reassembled structure is catalytically active, we carried out an enzymatic assay using a double-stranded DNA (dsDNA) as substrate (Fig. 2g). The results showed that the NF$_L$-CF$_L$ could digest DNA efficiently in the presence of Mg$^{2+}$, while neither NF$_L$ nor CF$_L$ alone exhibited any activity. When EsaG was supplemented into the reaction, the nuclease activity of NF$_L$-CF$_L$ was totally abolished. These results support that, during the secretion process by T7SS, the covalently linked NF$_L$ and CF$_L$ can reassemble into an active protein when EsaG is stripped off from EsaD-EsaG complex. In fact, many proteins can be split into fragments that reassemble into a functional protein spontaneously, such as ribonuclease S and green fluorescent proteins[32]. However, most of these fragments are generated through artificial design, and their reassembly and disassembly are not involved in the functional regulation of these proteins in the nature.

### Finger loop is the trigger of conformational changes on EsaDc

We next investigated how EsaG triggers the conformational changes of EsaDc. Notice that the EsaG binding surface of the CF$_L$ part of the free EsaDc is exposed to the solvent, while that of the NF$_L$ part is buried and stabilized by the ββα-metal finger motif (Fig. 1). It is possible that EsaG binds with the CF$_L$ first and triggers conformational change on the ββα-metal finger motif, leading to dislocation of NF$_L$ from CF$_L$. Structural superposition of the free and the EsaG-bound EsaDc through their CF$_L$ indicates steric clash between the finger loop on the free EsaDc and Loop$_{β4'-αD'}$ of EsaG in the complex (Supplementary Fig. 10).

Since the biological function of the finger loop remains unknown, we first explored its roles in substrate binding and DNA cleavage. Four residues (FKEK, 554–557) on the turn of the finger loop were truncated. The resulting constructs (EsaDc$^{-FKEK}$ and CF$_L^{-FKEK}$) exhibited similar expression level and behavior on chromatographic column to that of the full-length ones, suggesting that the truncation does not destroy the whole structure of the proteins. Electrophoretic mobility shift assay (EMSA) showed that EsaDc$^{-FKEK}$ retains strong DNA binding capacity (Supplementary Fig. 11a). Unlike the EsaDc-DNA interaction that could be total abolished by EsaG, EsaDc$^{-FKEK}$-DNA interaction still could be observed in the presence of EsaG, suggesting that the effect of EsaG to the ββα-metal finger motif was weakened by the truncation of finger loop. However, the DNase activity of NF$_L$-CF$_L^{-FKEK}$ complex was drastically reduced and could be further reduced by EsaG (Supplementary Fig. 11b). These results indicated that the finger loop is not an essential structural element for DNA binding, but might involve in the catalytic process, probably by stabilizing the catalytic center. In the free EsaDc structure, we indeed observed associations of the finger loop with residues surrounding the catalytic center: R552, F554 and K557 from the finger loop form a hydrophobic region with K548 and F549 located on the αB which embraces the Mg$^{2+}$ ion-coordinating residue N551; R552 further contributes two hydrogen bonds with the side chain of D525 from β4' (Fig. 3a).

We then carried out molecular dynamics (MD) simulations in combination of ratchet&pawl potential[33] to simulate the process of EsaG-EsaDc interaction. Starting from EsaDc and EsaG separated, the ratchet coordinate was defined as the RMSD to the structure of EsaDc-EsaG complex, and bias potentials were only damped when the system attempted to move in the opposite direction. No force was applied when EsaG was spontaneously moving towards EsaDc. During the process, it was noticed that the finger loop was twisted due to the approaching of Loop$_{β4'-αD'}$ on EsaG. F554 in the hydrophobic region moved inward to push R552, resulting the broken of the hydrogen bonds between R552 and D525. It is possible that the binding of $_{Loopβ4'-αD'}$ on EsaG triggered the conformational changes of the finger loop, which may deconstruct the hydrophobic and salt bridge interactions that maintain the structure of ββα-metal finger motif on EsaDc (Fig. 3a).

To further investigate the process of EsaG-EsaDc interaction, we carried out replica exchange with solute tempering (REST2) simulations[34]. The conformational space of EsaDc was sampled using 16 replicas with effective solute temperatures from 300 to 450 K. As a comparison, EsaDc in absence of EsaG was simulated under the same condition. The simulation time was set to 500 ns for each replica, resulting a total sampling time of 8 µs for EsaDc and EsaDc-EsaG complex, respectively. The conformation of EsaDc and its changes induced by EsaG were visualized using principal component analysis (PCA) based on backbone of EsaDc. As shown in Fig. 3b, c, the binding of EsaG significantly changed the conformational distributions of EsaDc. Several new populations emerged, indicating the occurrence of conformational heterogeneity upon binding to EsaG. The structures of the populations were examined and showed that although the NF$_L$ and CF$_L$ fragments did not separate in the time-limited sampling, the newly emerged structures of the two fragments were significantly more disordered. Especially, the ββα-metal finger motif in the presence of EsaG presented as highly flexible conformations. Together, the computational studies revealed that the binding of EsaG induce conformational changes of the finger loop first, which further leads to the disorder of the whole ββα-metal finger for the insertion mechanism.

### EsaDc-EsaG form heterotetramer in solution

Even though there is only one EsaDc-EsaG complex in the asymmetric unit, gel-filtration experiments showed that the complex was eluted at a volume corresponding to a molecular weight of 80 KD, close to that

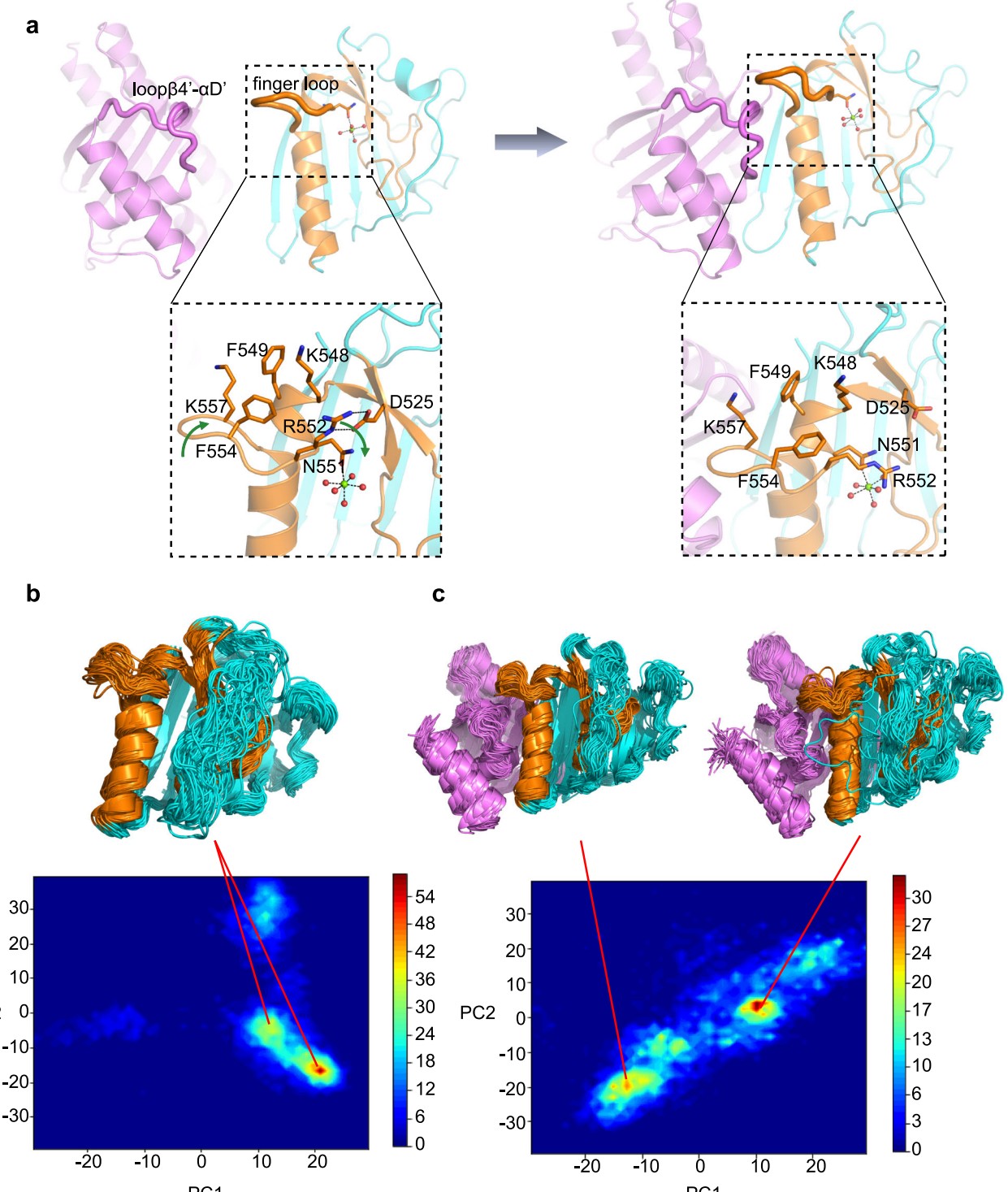

**Fig. 3 | Conformational changes of EsaDc in MD simulations. a** Conformational changes of finger loop during the binding process of EsaG to EsaDc. EsaG is colored in purple; the ββα-metal finger is colored in brown and the rest part of EsaDc in cyan. The residues on and around the finger loop are highlighted in expanded view with their side chains shown as stick presentation. The magnesium ion is shown as green sphere, and the coordinated waters are shown as red spheres. **b, c** PCA analysis for probability distribution of REST2 sampled EsaDc conformations in absence (**b**) or presence (**c**) of EsaG. The representative structures from high distributed popularity are shown in cartoon.

of two EsaDc-EsaG dimers (Supplementary Fig. 1). We further performed native mass spectrometry to investigate the presence of the tetramer state of EsaDc-EsaG complex. It is apparent that the oligomeric state of the EsaDc-EsaG complex exists and is concentration dependent (Fig. 4a). At low concentration (10 μM), the dominant oligomer in solution is EsaDc-EsaG dimer. When the concentration is

increased to 50 μM, the signals for tetramer are drastically strengthened.

To reveal how the EsaDc-EsaG tetramer is formed, we next searched the heterotetramer interface across the crystal units. There are two crystallographic packing contacts potentially involved in tetramer formation, both of which are generated by two of the same fragment of

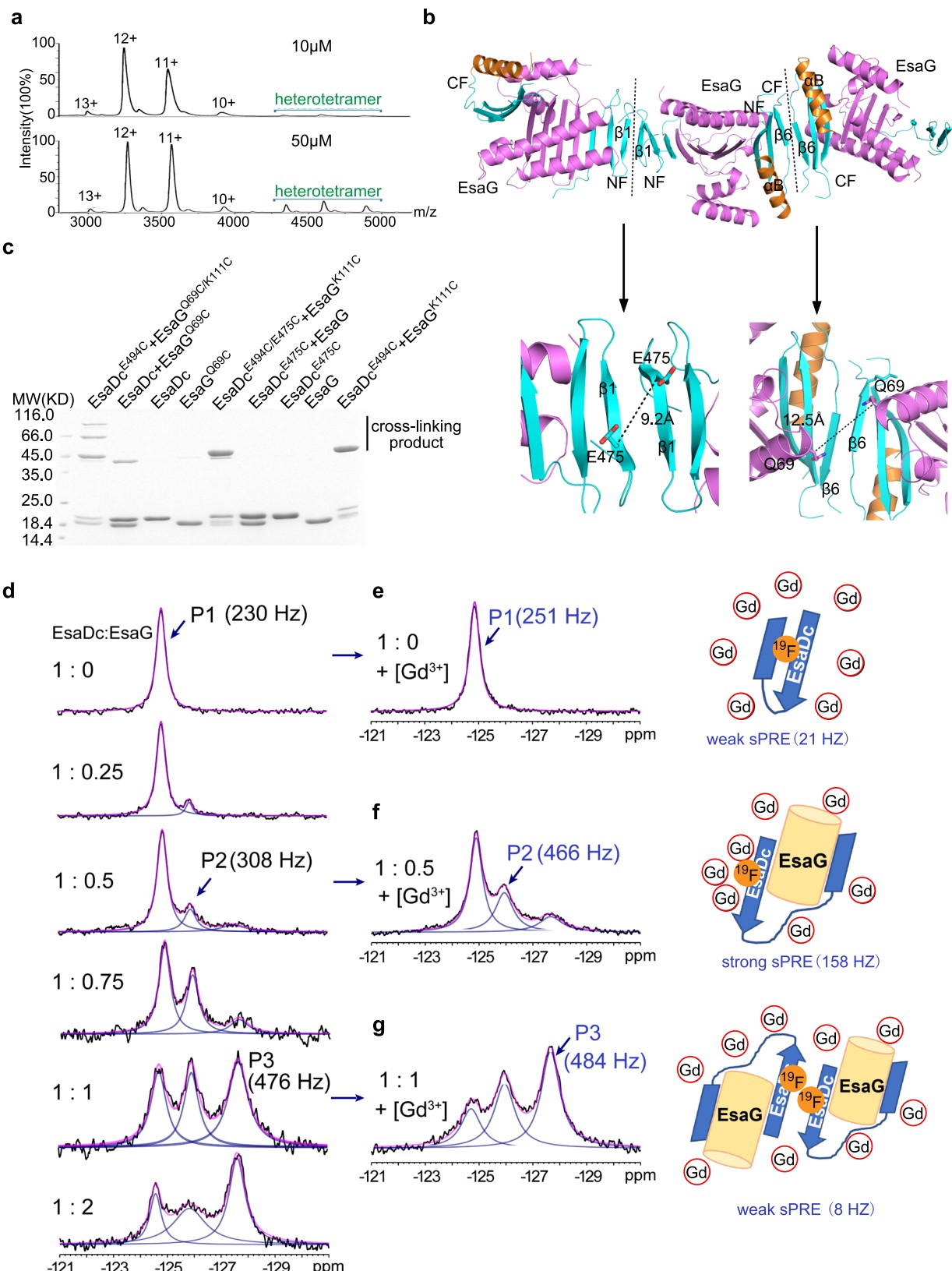

EsaDc in a 2-fold symmetry, while EsaG does not contribute any direct interaction (Fig. 4b). In the first contact, three-stranded β-sheets of two NFs interact through their anti-paralleled β1, while the second one is constructed by αB' and β6' from two CFs (Fig. 4b). To determine which interface mediates the tetramer formation, we evaluated the cross-linking of residue EsaDc[E475C], positioned at the NF-mediated interface,

and EsaG[Q69C], located at the CF-mediated interface, using IA probe (Fig. 4b). The results showed that strong signal of cross-linked product was generated only by EsaG[Q69C] in the presence of EsaDc, suggesting that two EsaDc-EsaG dimers form a tetramer through the CFs (Fig. 4c). We also combined these two mutations with the pair EsaG[K111C]-EsaDc[E494C] on the NF-EsaG interface, respectively. As expected, the

**Fig. 4 | EsaDc-EsaG may form tetramer through CF. a** Native mass spectrum of EsaDc-EsaG complex at low (10 μM) and high (50 μM) concentrations. Experimental molecular weights for dimer and tetramer calculated based on the assigned charge states are 39.092 KD (theoretical molecular weight 38.764 KD) and 77.789 KD (theoretical molecular weight 77.528 KD), respectively. **b** Ribbon representation of EsaDc-EsaG heterodimer interacting with nearby asymmetric units. The proteins are colored in the same scheme as shown in Fig. 1a. Potential dimer-to-dimer interfaces across the crystal unit cells are presented as black dash. Close-up view of the residues selected for cross-linking are highlighted below. **c** SDS-PAGE analysis of EsaG and EsaDc variants incubated with the IA probe. The experiments were performed twice with similar results. **d** $^{19}$F-NMR spectra of EsaDc-$^{19}$F-W567 in the presence of different equivalents of EsaG. Black lines are the experimental spectra, blue lines are deconvoluted peaks and magenta lines are the sum of deconvoluted peaks. **e**–**g** are the spectra (left) of EsaDc$^{19}$F-W567 mixed with the indicated equivalent of EsaG and 50 mM Ga-DTPA-BMA. The cartoon representations (right) are the solvent accessibility of the $^{19}$F-W567 in free EsaDc, corresponding to P1 (**e**), in EsaDc-EsaG dimer, corresponding to P2 (**f**), and in EsaDc-EsaG tetramer, corresponding to P3 (**g**), respectively. The sPRE strengths in Hz are indicated under the cartoons.

EsaG$^{Q69C/K111C}$- EsaDc$^{E494C}$ complex generated cross-linked products of trimer (~60 KD) and tetramer (~80 KD), while the EsaG$^{K111C}$-EsaDc$^{E494C/E475C}$ complex only exhibited a cross-linked dimer, as that of the pair EsaG$^{K111C}$- EsaDc$^{E494C}$ (Fig. 4c).

Interaction of EsaDc and EsaG and the interface for the tetramer formation were further confirmed by using solution fluorine NMR. There are two tryptophan residues in EsaDc (W560 and W567), both of which are buried in the free EsaDc structure, but exposed to solvent on the structure of the complex (Supplementary Fig. 12). The C5-position of W567 is pointing towards the solvent, making it a suitable target to be substituted with 5-fluoro-L-tryptophan (5F-L-Trp) and to study the conformational changes of EsaDc by $^{19}$F NMR. To simplify the NMR spectrum, W560 was mutated to phenylalanine. The resulted single site $^{19}$F labeled EsaDc (EsaDc-$^{19}$F-W567) showed a single peak (P1) at −124.8 ppm in the $^{19}$F NMR spectrum. Upon addition of EsaG, a new peak (P2) appeared at −125.9 ppm. With the increase of EsaG, a third peak (P3) appeared at −127.6 ppm and became the dominant peak (Fig. 4d). The sequential appearance of P2 and P3 indicated that the binding of EsaDc and EsaG is not a single process. Considering a significant change of the solvent accessibility of W567 based on crystal structures (Supplementary Fig. 12b), solvent paramagnetic resonance enhancement (sPRE) measurement[35] was performed to assess the solvent exposure of $^{19}$F-W567. Upon addition of a paramagnetic reagent gadodiamide (Gd-DTPA-BMA), which could broaden the resonance peaks for nucleic spins with close contact, the line width of P1 was only slightly changed, consistent with a buried $^{19}$F-W567 in EsaDc (Fig. 4e). In contrast, in the presence of EsaG, the newly appeared P2 showed a strong sPRE effect (158 Hz), indicating a solvent exposed environment of $^{19}$F-W567 which allows the paramagnetic Gd-DTPA-BMA approach with short distance. This is consistent with the crystal structure of the EsaDc-EsaG complex where W567 is exposed to solvent (Fig. 4f and Supplementary Fig. 12b). Surprisingly, P3, which is highly populated in the presence of high concentration of EsaG, did not show significant sPRE effect (Fig. 4g). We rationalize that P3 corresponds to the formation of EsaDc-EsaG tetramer, where W567 is located in a pocket comprised αB′ and β6′ from two CFs (Fig. 4f and Supplementary Fig. 12c). The pocket has Van der Waals diameter of <6 Å and is smaller than the paramagnetic reagent Gd-DTPA-BMA (vdW diameter: >8 Å)[36], thus, preventing a close contact between Gd-DTPA-BMA and $^{19}$F-W567 and resulting in a weak PRE effect. Overall, the sequential appearance of P2 and P3 as well as the strong sPRE for P2 and weak sPRE for P3 are consistent with the structural model of EsaDc-EsaG dimer and EsaDc-EsaG tetramer.

**A conserved inhibition mechanism in Gram-positive bacteria**

Sequence blast analysis identified more EsaG-like proteins in many other Gram-positive bacteria (Supplementary Fig. 13), sharing about 40% sequence similarity with EsaG. All these proteins are classified into a family named DUF600 (Domain of Unknown Function), due to limited knowledge about their biological function. One exception is YezG from *Bacillus subtilis*, one of the best characterized bacteria and usually used as a model organism for Gram-positive bacteria. Similar to EsaG, YezG interacts with and inhibits the activity of a nuclease domain located on the C-terminus of a toxin, YeeF[37]. To investigate whether YezG inhibits YeeF using a similar strategy, we cleaved the nuclease domain of YeeF into two fragments (YeeF$_{508-555}$ and YeeF$_{556-669}$) according to the sequences of NT$_L$ and CT$_L$ of EsaDc. As is the case for EsaD, the two fragments of YeeF could interact with YezG independently in the His-tag pull-down assay (Fig. 5a). They could also assemble into active form protein to digest dsDNA, but this activity is inhibited by YezG (Fig. 5b). These results imply that the inhibition of nuclease toxin through insertion of antitoxin into the core β-sheet of the toxin and deformation of the ββα-metal finger motif may be widely adopted among Gram-positive bacteria.

## Discussion

Due to their ubiquity among clinical isolates of pathogenic bacteria and the essential processes targeted, antimicrobial toxins are promising candidates for the development of strategies to combat bacterial infections[38], such as selectively removing strains carrying resistance genes to clinically relevant antibiotics. Comprehensive understandings of the biological function and the underlying molecular mechanism of these antimicrobial toxins are very important for this purpose. However, in contrast to the extensive and diverse examples of the toxic arsenal across Gram-negative bacteria, very few antibacterial toxins have been unraveled in Gram-positive bacteria[39]. EsaD is the first antibacterial toxin that was shown to be secreted through the T7SS of a Gram-positive bacterium. The studies here reveal mechanistic features of the EsaG-EsaD toxin-antitoxin system, which may pave the road for developing therapeutic approaches against *S. aureus* that target this antagonism.

The major contribution of this study is the discovery of an inhibitory mechanism in the toxin-antitoxin module that is based on a significant structural deformation of the toxin, where the central β-sheet of the nuclease domain of EsaD is split by EsaG into two fragments (Fig. 5c). The originally well-folded ββα-metal finger connecting the two fragments is stretched and becomes disordered when the N- and C-terminal fragments are set apart to sandwich EsaG, leading to the ruin of the catalytic site of EsaD and abolishment of the nuclease activity. Inspired by the structure of free EsaD and EsaDc-EsaG and supported by mutation and molecular dynamic simulations, we also proposed the important roles of the finger loop in the catalytic activity of EsaD and in the inhibition by EsaG. In addition to the EsaDc-EsaG dimer formation, a higher order tetramer formation in solution has also been proved by using native mass, cross linking and $^{19}$F-NMR. Since EsaG-like proteins exist in many other Gram-positive bacteria, this EsaD-EsaG-like toxin-antitoxin mechanism may be adopted widely.

The inhibitory mechanism disclosed in this study is clearly different from the well-studied IDR-dependent mechanism in most of the Type II toxin-antitoxin modules. Based on the structures of some Type II toxin-antitoxin systems, such as MazE-MazF[40], AtaR-AtaT[41], and CcdB-CcdA[42] from *E. coli* and VbhT-VbhA from *B. schoenbuchensis*[43], the antitoxins usually contain a DNA binding domain that is critical for transcriptional autoregulation and a toxin-neutralizing domain. The toxin-neutralizing domain is featured with an IDR that folds upon toxin binding to mask the substrate binding site or disrupt local structure around the catalytic site. In contrast, we didn't observe any disorder-to-order conversion of the antitoxin EsaG in the structures of EsaG and

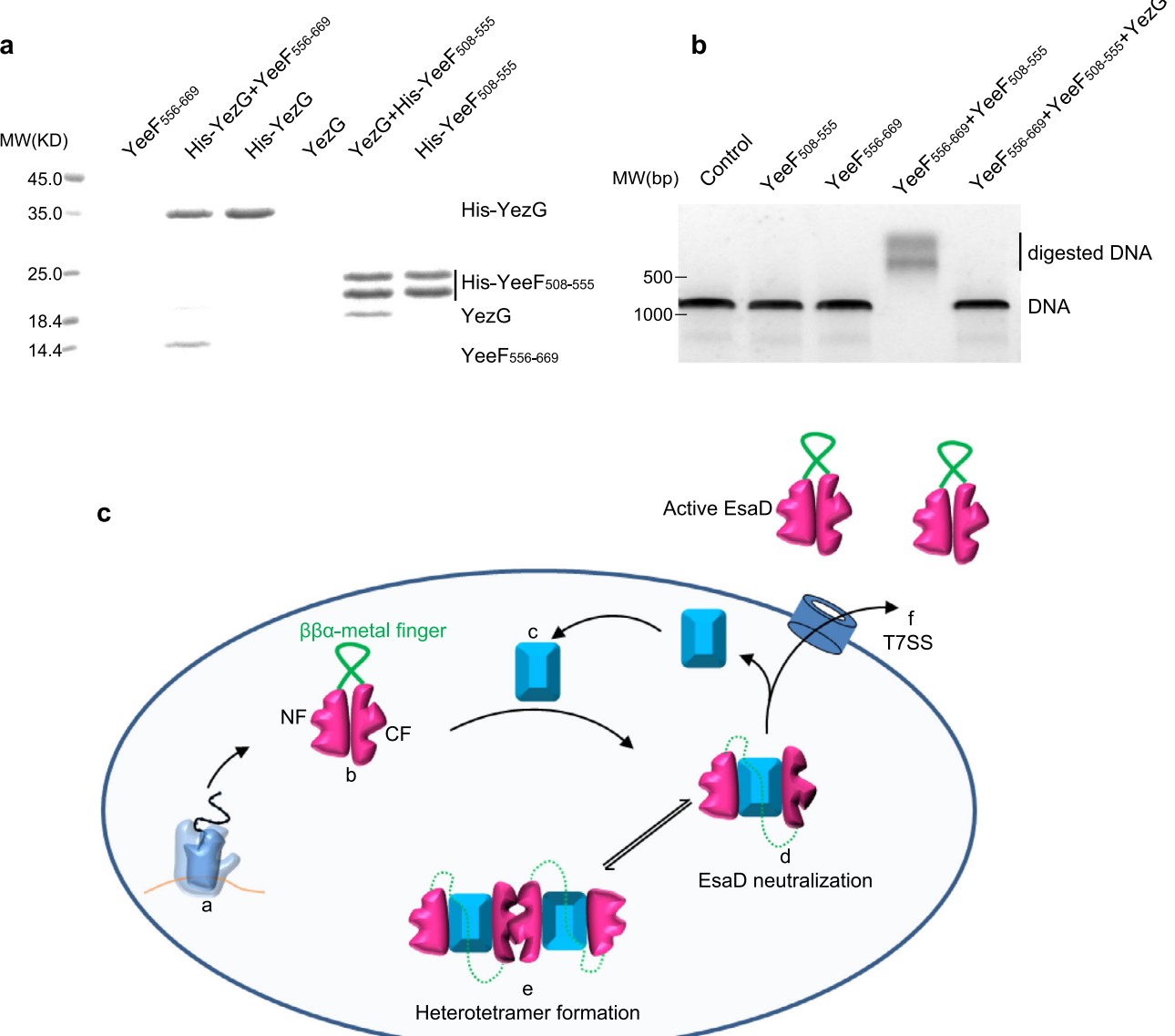

**Fig. 5 | A conserved toxin inhibition mechanism in Gram-positive bacteria.**
**a** His-tag pull down assay. A His-SUMO tag was co-expressed on the N-terminus of YezG (His-YezG) and YeeF$_{508-555}$ (His-YeeF$_{508-555}$) to chelate with the nickel beads. All the proteins were purified from *E. Coli*. Proteins are indicated by Coomassie blue staining. The experiments were performed twice with similar results. **b** In vitro nuclease activity of YeeF fragments. The gel was stained with ethidium bromide and visualized under UV light. The experiments were performed three times with similar results. **c** Cartoon representation of the molecular model for the mechanism of EsaD inhibition by EsaG. The nuclease activity of EsaD is inhibited upon its synthesis by insertion of EsaG monomer into the core structure of EsaD, leading to the disorder of the ββα-metal finger. The EsaD-EsaG complex may form tetramer when accumulated in the bacteria. EsaG proteins are removed from the complex and left behind when EsaD proteins are translocated through the T7SS. This model shows the ribosome (**a**), the newly synthetized EsaD (**b**), EsaG (**c**), EsaD-EsaG heterodimer (**d**), EsaD-EsaG heterotetramer (**e**) and the T7SS (**f**).

EsaDc-EsaG complex. Instead, the key of the inhibition lies on the order-to-disorder conversion of the substrate interaction site and catalytic site on the toxin. Of note, a recent study on RnlA-RnlB, a type II toxin-antitoxin system encoded by *E. coli* K-12 chromosome[44], also revealed complex conformational changes on the toxin RnlA[45]. The RnlA dimer can shift its domain orientation and dimerization interface to make it in equilibrium between an inactive resting state and an active state. The antitoxin RnlB only recognizes the active RnlA dimer and inhibits its activity by directly binding to the catalytic cleft of the RnlA dimer, a strategy used commonly in the Type II toxin-antitoxin systems. Thus, our antitoxin mechanism observed in EsaD-EsaG is also different to the RnlA-RnlB system.

Although the structures of EsaD and EsaG shed light on how the toxicity of EsaD is neutralized, more questions remain to be answered. For example, what is the biological relevance of the EsaD-EsaG

heterotetramer? The concentration of EsaD-EsaG complex has never been measured in the living bacteria, making it hard to assess the impact of the EsaD-EsaG heterotetramer. Since EsaD is secreted into surrounding environment to kill competitors, it is conceivable that high levels of EsaD are expressed and that the heterotetrameric arrangement of EsaD-EsaG may help keep EsaD in the inhibited state before secretion.

Another immediate question is how EsaG is removed from the complex by the T7SS. Given the large binding surface between EsaDc and EsaG, this might be a very energy-consuming process, possibly allosterically modulated by other components in the T7SS. Overall, our studies on EsaD and its counterpart EsaG add to our understanding of the toxin-antitoxin system in Gram-positive bacteria and provide the structural basis for developing therapeutic approaches that target this antagonism.

## Methods

### Expression and purification

Genes encoding EsaG, EsaDc, $NT_L$ and $CT_L$ were cloned into a modified pRSF-Duet vector which is generated by inserting the sequence of SUMO-tag ahead the multiple cloning site, respectively. The proteins were over-expressed with a fused N-terminal His$_6$-SUMO tag in *E. coli* BL21(DE3) cells. Cells were induced with 0.4 mM isopropyl β-D-1-thiogalactopyranoside (IPTG) at an optical density at 600 nm of 0.8 and grown at 16 °C overnight. The fusion proteins were first purified using a nickel column. The His$_6$-SUMO tag was removed by ULP1. The tagless proteins were further purified using ion-exchange column (GE Healthcare) and the Superdex 200 size exclusion column (GE Healthcare). For NMR study, the $^{15}$N-labeled proteins were produced in M9 minimal media supplemented with $^{15}$NH$_4$Cl as sole nitrogen resource, and the $^{19}$F-labeled EsaDc was produced in M9 minimal media supplemented with 120 mg/L of 5-fluoro-D/L-tryptophan.

### Crystallization and structure determination

Crystallization experiments were carried out using hanging-drop diffusion method. The crystals of EsaG were grown in a buffer containing 0.1 M sodium phosphate/citric acid (pH 4.2), 10% 3350 and 0.3 M NaCl. The crystals of the nuclease domain of EsaD were grown in a buffer containing 0.2 M Mg(CH$_3$COO)$_2$ and 20% PEG3350. The crystals of EsaDc-EsaG were grown in a buffer containing 0.1 M Tris (pH 8.5), 12% PEG6000, 0.15 M NaCl, 10% Ethylene glycol, 0.5% Tacsimate (pH 7.0) and 1.0% PEG5000MME. Crystals were cryo-protected using the precipitant solution supplemented with glycerol or ethylene glycol to a final concentration of 20–30% (v/v) before flash frozen in liquid nitrogen. Datasets were collected on the beam-line BL02U1, BL17B1 and BL19U at the Shanghai Synchrotron Radiation Facility. The data were indexed, integrated and scaled by the HKL2000 package[46] and Aquarium[47]. The structure of EsaG was obtained by Se-SAD. The structure of EsaDc was solved by molecular replacement using a model generated by AlphaFold[48]. Iterative rounds of model building in COOT[49] and refinement in Phenix[50] were carried out. Data collection and refinement statistics are summarized in Supplementary Table 1.

### His-tag pull-down assay

All proteins used in this assay were expressed in *E. coli* and purified to high purity. The pull-down assays were performed by incubating appropriate amount of Ni-NTA beads, 100 μL of protein mixtures (His-Sumo tagged proteins were mixed with un-tagged proteins with a molar ratio of 0.16 mM: 0.2 mM) and 900 μL of binding buffer (25 mM Tris-HCl, pH 8.0 and 1 M NaCl) at 24 °C for 30 min. The Ni-NTA columns were then washed three times with washing buffer (25 mM Tris-HCl, pH 8.0, 1 M NaCl and 25 mM imidazole) to remove unbound proteins. Finally, the bound proteins were eluted with 100 μL elution buffer containing 25 mM Tris-HCl, pH 8.0, 1 M NaCl and 250 mM imidazole. The eluted proteins were analyzed by using SDS-PAGE.

### In vitro DNase assay

The nuclease activity of EsaDc was tested using double-stranded DNA as substrate. Briefly, the $CF_L$ and $NF_L$ were mixed in equal proportion (0.12 μM) to form complex in reaction buffer (25 mM Tris-HCl, pH 8.0, 200 mM NaCl and 2 mM MgCl$_2$). 0.5 ng DNA were supplemented into 10 μL $CF_L$- $NF_L$ mixture to start the reaction at room temperature for 30 min. After that, the samples were analyzed on a 1.2% agarose gel. The gel was stained by ethidium bromide and visualized with a UV transilluminator.

### Iodoacetamide probe synthesis

2-iodoacetic acid (186.0 mg, 1.0 mmol) and 1,5-pentanediaminewere (112.4 mg, 1.1 mmol) were dissolved in 5 mL dichloromethane (DCM) in a dry flask with continuous stirring, followed by addition of 3-(3-dimethylaminopropyl)−1-ethylcarbodiimide hydrochloride (EDCI, 287.6 mg, 1.5 mmol). When the temperature of the reaction mixture was cooled to 0 °C, 4- dimethylaminopyridine (DMAP, 12.2 mg, 0.1 mmol) was added dropwise and reacted for 1 h at 0 °C. The mixture was then warmed up to room temperature and stirred for another 24 h. The reaction was quenched by addition of 5 mL saturated NaHCO$_3$ and the product was extracted with dichloromethane (2 × 10 mL). The extracted organic layers were dried by anhydrous Na$_2$SO$_4$, and the crude product was further purified by silica gel flash column chromatography (DCM/MeOH = 50:1) with a total yield of 56%. $^1$H NMR (300 MHz, MeOD) δ 4.03 (s, 4H), 3.25 (t, J = 6.5 Hz, 4H), 1.63–1.49 (m, 4H). $^{13}$C NMR (101 MHz, MeOD) δ 169.30, 43.16, 40.36, 27.54. HRMS (ESI) calcd for C8H15I2N2O2, 424.9217 [M + H]$^+$; found, 424.9208.

### Cross-linking and mass spectrometry

0.6 ng protein (final concentration: 75 nM) was mixed with 0.3 μM IA probe in a buffer containing 25 mM Tris-HCl (pH 8.0), 200 mM NaCl and 2 mM MgCl$_2$ with a final volume of 20 μL. The reaction was performed at room temperature for 1 h in the dark. Afterward, the reaction products were subjected to SDS-PAGE and the band of the cross-linked proteins were recovered, excised into small pieces and destained by 50 mM NH$_4$HCO$_3$ in 50% acetonitrile. The gel pieces were washed with 100% acetonitrile, and then dried using a SpeedVac. Finally, the samples were digested with 12.5 ng/μL trypsin (Promega) in 50 mM NH$_4$HCO$_3$ for 16 h at 37 °C. 50% acetonitrile/5% formic acid was added to stop the reaction and extract the digested peptides. The solution was dried out and desalted using a Mono-Spin C18 column (GL Science, Tokyo, Japan).

The peptide mixture was analyzed by a home-made 30 cm-long pulled-tip analytical column (75 μm ID packed with ReproSil-Pur C18-AQ 1.9 μm resin, Dr. Maisch GmbH), the column was then placed in-line with an Easy-nLC 1200 nano HPLC (Thermo Scientific) for mass spectrometry analysis. The analytical column temperature was set at 55 °C during the experiments. The mobile phase and elution gradient used for peptide separation were as follows: 0.1% formic acid in water as buffer A and 0.1% formic acid in 80% acetonitrile as buffer B, 0–1 min, 3–6% B; 1–96 min, 6–36% B; 96–107 min, 36–60% B, 107–108 min, 60–100% B, 108–120 min, 100% B. The flow rate was set as 300 nL/min.

Data-dependent MS/MS analysis was performed with a Q Exactive Orbitrap mass spectrometer (Thermo Scientific). Peptides eluted from the LC column were directly electrosprayed into the mass spectrometer with the application of a distal 2.5-kV spray voltage. A cycle of one full-scan MS spectrum (m/z 300–1800) was acquired followed by top 20 MS/MS events, sequentially generated on the first to the twentieth most intense ions selected from the full MS spectrum at a 28% normalized collision energy. Full scan resolution was set to 70,000 with automated gain control (AGC) target of 3e. MS/MS scan resolution was set to 17,500 with isolation window of 1.8 m/z and AGC target of 1e. The number of microscans was one for both MS and MS/MS scans and the maximum ion injection time was 50 and 100 ms, respectively. The dynamic exclusion settings were as follows: charge exclusion, 1 and >8; exclude isotopes, on; and exclusion duration, 5, 10, 15 s, respectively. MS scan functions and LC solvent gradients were controlled by the Xcalibur data system (Thermo Scientific).

Peptides were identified using pLink2 software (version 2.3.9, pFind Team, Beijing, China)[51]. The sequences of EsaD and EsaG were used for cross-linked peptides search and the mutated residue was replaced by cysteine. Search parameters in pLink were: enzyme: trypsin; missed cleavages: 3; precursor and fragment tolerance: 20 ppm; minimum peptide length: 6. Oxidation of methionine was set as variable modification. The results were filtered by applying a 5% FDR cutoff at the spectral level.

### Native mass spectrometry

The protein was dialyzed overnight at 4 °C against a buffer containing 100 mM CH$_3$COONH$_4$. Then protein complex solution (3 μL, desalted)

was analyzed with a quadrupole ion mobility time-of-flight mass spectrometer (SYNAPT G2-Si HDMS, Waters, USA) in positive ion mode, respectively. Protein ion was generated by nano-ESI from a homemade borosilicate capillary emitter that pulled by using a P-97 puller (Sutter Instruments, Novato, CA, USA) and electrically connected to high voltage with a platinum wire. The instrumental parameters were as follows: capillary voltage 0.8–1.2 kV, sampling cone 40 V, source offset 80 V, and source temperature 37 °C. The data was processed by Masslynx 4.1(Waters, USA).

### Analytical ultracentrifugation (AUC)
Sedimentation velocity experiments were performed in a Beckman Optima AUC-A/I analytical ultracentrifuge using a UV/VIS optical system at a wavelength of 280 nm. The sample was applied in a concentration of 0.2 mg/mL in a buffer containing 25 mM Tris-HCl, 100 mM NaCl, 5 mM DTT, pH 8.0. The sedimentation coefficient (s) and distribution [c(s)] were determined from sedimentation speed experiments at 42,000 rpm for overnight at 20 °C. Data analysis was performed in Sedfit[52].

### Biolayer interferometry (BLI)
An OKTET K2 system (ForteBio Inc., Menlo Park, CA, USA) and streptavidin (SA)-coated biosensors were used to measure association and dissociation kinetics for Avi-tagged EsaG in relation to EsaD$_C$. Before the experiment, the biosensors were equilibrated in a buffer (300 mM NaCl, 30 mM NaH$_2$PO$_4$, 27 mM KCl, pH7.2) for 10 min, then the biotinylated EsaG was immobilized on streptavidin biosensors. The biosensors were then exposed to different concentrations of EsaDc. Data were analyzed and the binding curves were fit using the Data Analysis 10.0 software package (ForteBio). Association and dissociation curves were exported using Excel format and imported into Prism (GraphPad software) for visualization of the sensorgram.

### NMR experiments
NMR experiments were performed on a Bruker Advance III HD 600-MHz spectrometer equipped with a BBO BBF-H-D probe at 20 °C. Proteins were dissolved in a buffer containing 200 mM NaCl, 27 mM KCl, 30 mM Na$_2$HPO$_4$, 1.8 mM KH$_2$PO$_4$, 10% D$_2$O, and at pH 6.5. Two-dimensional $^1$H,$^{15}$N-HSQC (heteronuclear single quantum coherence) spectra were collected with a spectral width of 30 ppm in the $^{15}$N dimension and 18 ppm in the $^1$H dimension, with a recovery delay of 1.0 s. Specifically, for $^{15}$N-EsaDc (0.4 mM), the spectra were acquired with 1024 complex points in direct and 128 complex points in indirect dimensions for $^{15}$N-EsaDc and 224 scans per increment. For $^{15}$N-CF$_L$ (0.25 mM) in the presence of unlabeled NF$_L$ (1.0 mM), the spectra were acquired with 1024 complex points in direct and 192 complex points in indirect dimensions and 320 scans per increment. For $^{15}$N-NF$_L$ (1.0 mM) in the presence of unlabeled CF$_L$ (1.5 mM), the spectra were acquired with 1024 complex points in direct and 192 complex points in indirect dimensions and 112 scans per increment. All the spectra were collected with a spectral width of 30 ppm in the $^{15}$N dimension and 18 ppm in the $^1$H dimension, with a recovery delay of 1.0 s. NMR data were processed with TOPSPIN (Bruker) and analyzed with SPARKY (Goddard and Kneller, SPARKY 3).

$^{19}$F NMR spectroscopy was conducted on a Bruker Advance III HD 600 MHz spectrometer equipped with a TCI H&F/C/N-D probe at 25 °C. 1D $^{19}$F-NMR spectra were recorded using the standard ZG pulse in the Bruker pulse library, with the TD points and the spectral width set to achieve an acquisition time of about 100 ms. The carrier frequency was set at −120 ppm. The recycle delay was set to 1 s. All the spectra were recorded at 25 °C without decoupling. EsaDc-$^{19}$F-W567 protein samples were prepared at 160 μM in buffer containing 300 mM NaCl, 25 mM Tris-HCl (pH8.0), 2 mM MgCl$_2$, 10% D$_2$O. Unlabeled EsaG protein (3 mM in the sample buffer) was added into EsaDc-19F-W567 sample to achieve desired ration. For the sPRE measurement, Gd-DTPA-BMA/H$_2$O stock solution was added to a final concentration of 50 mM. All $^{19}$F NMR spectra were processed using MestRaNova 12.0.0 (Mestrelab Research) employing a 30 Hz exponential windows function. Spectra were baseline-corrected and the peaks were fitted using Lorentzian peak shapes. The sPRE values were calculated by subtracting the original linewidth from the peak linewidth in the presence of Gd-DTPA-BMA.

### Molecular dynamics simulation
The systems were built in Maestro and prepared by Protein Preparation Wizard (Schrödinger, LLC, New York, NY, 2020). The simulations were carried out using GROMACS 2020 patched with open-source, community-developed PLUMED library, version 2.6[53,54] equipped with CHARMM36m[55] force field for the proteins and ions, and TIP3P model for water. Energy minimization and restrained equilibration simulations in the NVT ensemble (T = 300 K) and NPT ensemble (T = 300 K, P = 1 atm) were conducted before production runs. During the simulation, a time step of 2 fs was used and LINCS algorithm was applied to the bonds involving hydrogen atoms. The cutoffs for the nonbonded interactions were 1.0 nm and the particle mesh Ewald method was applied. For the binding process of EsaG to EsaDc, 500 ns simulations were conducted with ratchet&pawl potential[33]. The potential was defined as

$$V(\rho(t)) = \begin{cases} \frac{k}{2}\left(\rho(t) - \rho_m(t)\right)^2, \rho(t) > \rho_m(t) \\ 0, \rho(t) \leq \rho_m(t) \end{cases} \quad (1)$$

with

$$\rho(t) = \left(s(t) - s_{target}\right)^2 \quad (2)$$

and

$$\rho_m = \min_{\tau \in [0,t]} \rho(\tau) + \eta(t), \quad (3)$$

where is $\eta(t)$ an additional fluctuation acting on the minimum value of $\rho(t)$ during the simulation and is the ratcheting coordinate defined as the root-mean-square deviation (RMSD) to the EsaDc-EsaG complex. A small force constant $k = 0.05 \, kcal/mol/^2$ was used. And the potential was zero when EsaDc and EsaG moving towards each other and damped the fluctuation when they moved oppositely. The last frame from ratchet&pawl simulation was used as the starting conformation for REST2 simulations[34] and replicated for 16 times with effective temperature spaced exponentially between 300 K and 450 K for residues from S444 to G558. The reason we chose this part but not the full length EsaDc for REST2 simulations is that EsaG interacts with CF (residue 559–614) of EsaDc first to trigger conformational changes in the NF and the ββα-metal finger motif (residue 444-558). And during the interaction, CF undergoes no obvious conformational change. The simulations were performed for 500 ns for each replica, resulting total simulation times of 8 μs for EsaDc and EsaG-EsaDc complex. Exchange between neighboring was attempted every 2 ps. The averaged exchange acceptance ratio for EsaDc and EsaG-EsaDc were 37.3% and 20.0%, respectively. Principal component analysis (PCA) was performed for the backbone of EsaD using MDAnalysis[56].

### Electrophoretic mobility shift assay (EMSA)
The DNA binding was analyzed by electrophoretic mobility shift assays. The in vitro binding reactions contained EsaDc (50 μM) and vary concentrations of EsaG (6.25–50 μM) were mixed with 90 ng of dsDNA (PCR product) in 10 μL reaction buffer (25 mM Tris, pH 7.5, 100 mM NaCl, 2 mM MgCl$_2$) followed by incubation at room temperature for 30 min. The samples were resolved on 1% agarose gel at

4 °C for 60 min at 70 V, stained with ethidium bromide and visualized using gel documentation system.

**Reporting summary**

Further information on research design is available in the Nature Research Reporting Summary linked to this article.

## Data availability

The atomic coordinates and structural factors for EsaG, the nuclease domain of EsaD and EsaG-EsaDc complex have been deposited into the PDB under the accession code 8GUN, 8GUP, and 8GUO, respectively. The structure of BH3703 has been published in PDB under the accession code 3IOT. The coordinates of the initial and final models of EsaDc-EsaG complex from the MD simulations are provided as Supplementary Data 2. Source data are provided with this paper.

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

## Acknowledgements

We thank Dr. Jikui Song for suggestive comments on our manuscript. This work was supported by funds from National Natural Science Foundation of China (32171184 to Z.-M.Z., 81922062 to X.L.), National Science Fund for Distinguished Young Scholars of China (31725022 to Y.W.), Key Program of Natural Science Foundation of China (31930084 to Y.W.), Guangdong Province (2019QN01Y979, 2022A1515012266) and Guangzhou City (202102010044) to Z.-M.Z. G.L. is supported by the Science Foundation of Guangdong Province of China (2019A1515110698), and Traditional Chinese Medicine Project of of Guangdong Province (20202038). We thank the staffs from BL17B/BL18U1/BL19U1/BL19U2/BL01B beamline of National Facility for Protein Science in Shanghai (NFPS) and BL02U1 at Shanghai Synchrotron Radiation Facility, for assistance during data collection. We also thank Dr. Chao Peng and Weiqian Wang of the Mass Spectrometry System at the National Facility for Protein Science in Shanghai (NFPS), Shanghai Advanced Research Institute, Chinese Academy of Science, China for data collection and analysis.

## Author contributions

W.Yongjin, J.L., L.D., H.H., L.D., Z.X. W.Yonghua, and Z.-M.Z. purified the proteins, crystalized the proteins and determined the structures. C.S., Y.H., and G.L. designed and performed the NMR experiments. W.Yongjin and H.H. performed pull-down and in vitro enzymatic assay. S.L. and Z.L. synthesized the probe. W.Yongjin and H.Z. designed and performed cross-linking assay. X.L. and Y.Z. performed computational study. P.S., K.D., and Z.-M.Z. conceived and oversaw the project.

## Competing interests

The authors declare no competing interests.
