## [Peer Review File · Nature Communications]

REVIEWER COMMENTS

Reviewer #1 (Remarks to the Author):

This manuscript deals with the crystal structures of the catalytic domain of the EsaD toxin and its complex with the EsaG antitoxin. The antitoxin disrupts the EsaD dimer by breaking its main beta-sheet, leading to a conformational change that prevents its nuclease activity. A model is presented that explains regulation of the activity via a series of distinct steps and involving 4 or 5 distinct species.

While this is an interesting finding, little supporting information is presented that would strengthen the conclusions and reject alternative hypotheses. Furthermore, the paper is plagued by poor and inaccurate descriptions, that make it difficult to understand what exactly happens and whether the data are interpreted correctly. Although the last (schematic) figure is revealing in this respect (they are not numbered in the document), the text itself is often confusing, in part because of poor English and the use of incorrect terminology (for examples see below). In absence of the crystal structures (which were not made available for refereeing) it is always difficult to establish if the conclusions drawn from a structure are likely correct or not (the validation reports say something about the quality of the structure itself, but give no clue if it is correctly interpreted in terms of biology). Missing is validation of the structures using SAXS given that true interfaces of the complexes need to be distinguished from lattice contacts.

The conformational changes that are reported, are certainly complex and it is unlikely that these could be predicted. However, there is at least one other TA system where (other but) equally large changes in the toxin are observed upon binding the antitoxin, and also in that case an alternative mode of dimer formation of the toxin is observed: *E. coli* rnlAB (Garcia-Rodriguez et al, 2021, NAR 49(12):7164-7178). This work is not cited nor discussed as it should be.

Some linguistic and other issues:

line 14: "once" -> "originally"

line 32: "complicated" -> "complicate" (in the past tense it means the problem is now already solved).

line 60: "EsaD was reported to be positioned in the membrane". It is not clear if the membrane of *S. aureus* is meant, or the membrane of a target cell

line 67: "with the C-terminal nuclease domain and neutralize the toxicity of EsaD" -> with the C-terminal nuclease domain of EsaD and neutralize its toxicity.

lines 80-81: "However, secondary structure prediction indicated neither intrinsic disordered region nor DNA binding domain in EsaG"

While the crystal structures confirm these statements, it should be noted that this cannot be derived from prediction. For protein segments that fold upon binding, both secondary structure predictors based on sequence as well as Alphafold2 tends to predict the folded state. Also disorder predictors do not recognize folding-upon)binding IDP domains of antitoxins as unfolded.

lines 94-95: "we launched a systematic effort to solve the structures ..." > "we determined the structures ...". Solving three structures is not a systematic effort.

line 95: "of the nuclease domain" -> "of the nuclease domain of EsaD"

line 111: "The conformation of EsaDc is out of expectation in the EsaDc-EsaG complex structure". This is incorrect English. Furthermore, one should not talk about a "complex structure" (which means that the structure is very complicated" but about "the structure of the complex". Please correct this throughout the text.

line 112: "asymmetric unit cell" -> "asymmetric unit" (an "asymmetric unit cell does not exist", you talk either about the asymmetric unit or about the unit cell, which are two distinct things that should not be mixed)

line 119: "the central 6-stranded beta sheet of EsaDc is split in two separate subunits". It is NOT split in SUBUNITS (that is a wrong term here). You should just write that "it is split".

line 121: "from the right side of $\beta 4$ ". There is no such thing as a right or a left side. Nor a top and bottom or back and front. This depends on the orientation of your molecule, and there is no universal reference frame. Therefore, this type of description is meaningless.

line 136: "All the secondary structures" -> "All the secondary structure elements"

line 163: you either "performed A His-tag pull down assay" or you "performed His-tag pull down assayS"

Same line 170: "We also performed A cross-linking assay" or "We also performed cross-linking assayS" depending on whether there was one or more than one different cross-linking assay

Line 168: I do not understand how a pull-down assay can tell you anything about the stoichiometry and conformational changes that occur upon interaction between EsaD and EsaG. The cross-linking mass spec experiments are more revealing in this respect, but to be really convincing, SAXS analysis of free proteins and the complex need to be performed as well.

Lines 184-186: "Another discovery in the pull-down assay is the association between the NT and the CT (Fig 2d), we thus deduced that these two fragments may reassemble into an intact structure of EsaDc in the absence of EsaG"

This is not really surprising as it is the case for many proteins to produce a correctly folded structure out of two fragments. For example, a number of applications using GFP in cell biology are based on this property. Spending a whole paragraph on this is therefore not warranted.

lines 205-206: What do you mean with "is well preserved in the crystal lattice"? Do you mean the asymmetric unit contains a monomer and the dimer is formed according to crystal symmetry? If so, rephrase correctly. If you mean something else, also rephrase so the reader knows what you mean. In any case, confirmation of the dimer arrangement via SAXS is required.

line 220: "Each of the EsaG structure in the dimer ..." This reads like in the dimer, the two EsaG monomers adopt different conformations. I think you mean "Each of the EsaG monomers in the dimer ..."

lines 223-224: "However, severe steric clash occurs between Subunit1 and the unoverlapped EsaG in the homodimer (Fig S9)"

I do not understand this sentence. What is "Subunit1"? It reads like in your crystal structure the two EsaG monomers that make up the EsaG dimers clash with each other, which is a contradiction.

Reviewer #2 (Remarks to the Author):

This manuscript presents a structure function study of a toxin antitoxin complex from the staphylococcal T7SSb. The authors provide an in-depth understanding of the molecular mechanism, by which the antitoxin EsaG inhibits the nuclease toxin EsaD. It is a very nice study which makes an important contribution to the T7SS field.

Comments:

-the authors should briefly explain why they used a truncated version of EsaD for their crystallization studies and subsequently for all functional assays. Does the proposed inhibition mechanism also work with full-length EsaD?

- In the prey cell EsaG is supposed to neutralize EsaD. How fast are EsaG-EsaDc complexes formed and what is the association rate?

- in the cytoplasm the EsaD-EsaG complex is bound by EsaE. Could this have an effect on complex formation/inhibition?

-The proposed heterotetramer formation is interesting but does not contribute to the inhibition mechanism itself. Could the authors speculate on its potential biological relevance in the discussion?

- Pull down assay: it is unclear to me what was loaded onto the NiNTA column: purified proteins or cell lysates containing the overexpressed proteins. The figure legend for 2d) could use a description.

- the Ramachandran Plot statistics should also be provided in table 1.

-the labeling is a bit inconsistent and could also be clearer. Subunit 1/2 would be better labeled as EsaDc-Subunit1/2 or perhaps better EsaDc subdomain 1/2. Pull-down assay: Perhaps use His-EsaDc-Subdomain 1/2 instead of His-Sumo-Nt/Ct as label.

-the sentence in lines 79-82 should be reformulated. This sounds like as if secondary structure predictions can be used to predict DNA binding sites.

-spelling errors: line 71, therapeutic; line 174: Å; line 286: subunit1.

Reviewer #3 (Remarks to the Author):

In the presented paper, the authors report the structure of EsaD separate, and in complex with EsaG. It is shown that significant structural rearrangements of the EsaD's c-terminal occur upon binding to EsaG. Although the results on the interactions between EsaD and EsaG are interesting, I don't think they are significant enough for Nature communications. Therefore, the paper would be a better fit in a more specialized journal. Finally, I have some comments below that should be addressed before publication.

1. The authors only crystallize the c-terminal of EsaD with and without EsaG. They should motivate how we can be sure that these structures are representative for the full EsaD-EsaG complex.

2. The statements about the polymerization states of EsaG and EsaD through the paper can be a bit confusing. In the introduction the authors say that in the free state EsaD is a monomer and EsaG is a dimer. But in the bound state, EsaD is a dimer and EsaG is a monomer, therefore the complex should be a heterotrimer right? This is not consistent with the results paper of the section. For example in "Dimer to monomer conversion of of EsaG during EsaDc-EsaG interaction" the authors say that the EsaDc-EsaG is a heterotetramer according to gel-filtration experiments and that two EsaDc-EsaG complexes might interact. I believe the authors suggest an EsaG-EsaD-EsaD-EsaG arrangement, but then it should have been made clear in the introduction.

Comments on MD simulation:

3. The error bars for the energy profiles from Umbrella sampling look really large (Figure 4C). I suspect this is because the simulation length for each window are really short, only 5 ns. They should probably be around 10-15 ns to improve the errors.

4. The authors state that bootstrap analysis was used to calculate the errors, but they need to give more details. There are several approaches to do this for umbrella sampling, and some can underestimate the errors. See for example this publication <https://pubs.acs.org/doi/pdf/10.1021/ct100494z>.

5. The authors say that equilibration simulations were done at the temperature of 277 K and don't say anything about the temperature in the umbrella sampling simulations. What was the temperature in these simulations? If the temperature in the umbrella sampling simulations was also 277 K the authors should motivate this choice, considering the structural changes are expected to occur at room temperature in vivo.

6. The authors should report the force constant that was used for window restraints in the umbrella sampling simulations.

7. Finally, the CHARMM36 protein force field have been upgraded to CHARMM36m in 2017 and should be used in future simulations.

We thank all the reviewers for their critical comments on our work. We tried our best to address all their concerns and revised the manuscript accordingly. Please find our point-to-point response to each of the reviewers' comments below.

REVIEWER COMMENTS

Reviewer #1 (Remarks to the Author):

While this is an interesting finding, little supporting information is presented that would strengthen the conclusions and reject alternative hypotheses. Furthermore, the paper is plagued by poor and inaccurate descriptions, that make it difficult to understand what exactly happens and whether the data are interpreted correctly. Although the last (schematic) figure is revealing in this respect (they are not numbered in the document), the text itself is often confusing, in part because of poor English and the use of incorrect terminology (for examples see below). In absence of the crystal structures (which were not made available for refereeing) it is always difficult to establish if the conclusions drawn from a structure are likely correct or not (the validation reports say something about the quality of the structure itself, but give no clue if it is correctly interpreted in terms of biology). Missing is validation of the structures using SAXS given that true interfaces of the complexes need to be distinguished from lattice contacts.

Response: We greatly appreciate the reviewer's constructive suggestions and critical comments on our manuscript, and also realize the inefficiency of supporting information to reach the conclusions. We have included the following work in the revised manuscript:

- 1) We performed different assays to investigate the oligomer state of EsaG. Even though the crystal structure of EsaG and gel-filtration analysis imply a dimer state, other experiments including chemical cross-linking and density gradient centrifugation support a monomer form. Actually, we also solved the structure of another EsaG-like protein, YezG, which is an antitoxin to inhibit the nuclease activity of the toxin YeeF in the Gram-positive bacterium *Bacillus subtilis*. YezG shares about 40% sequence identity with EsaG. It is also eluted from gel-filtration column at a dimer-size position. However, not dermic interface was found in the YezG structure (Fig. 1 in next page). The different packing patterns of EsaG, YezG and BH3703 further indicate that EsaG should be a monomer in solution. We also tried to follow the review's suggestion to perform SAXS at Shanghai Synchrotron Radiation Facility, but our appointment was postponed indefinitely because of the**

Fig. 1 The packing pattern of YezG in the crystal.

The YezG structure is not included in this paper.

outbreak of COVID-19 in Shanghai. We are really sorry for that, and hope our alternative experiments could satisfy the reviewer.

- 2) For the EsaDc-EsaG tetramer, we performed native mass spectrum, cross-linking and ¹⁹F-NMR studies. The results confirm the existence of tetramer state of EsaDc-EsaG complex in solution. Additionally, the cross-linking assay and ¹⁹F-NMR studies revealed that it is the CFs that involve in the tetramer formation, but not the NFs. We have revised our model accordingly.**
- 3) We performed functional studies on the finger loop of EsaG and MD studies on the process of EsaDc-EsaG interaction, which together revealed that the finger loop is the trigger of the conformational changes on EsaDc induced by EsaG.**
- 4) We performed His-tag pull-down and in vitro enzymatic assays on another toxin, YeeF, to show that YeeF could also be split into two separated fragments, which can interact with YezG independently and reassemble into a functional protein in the absence of YezG. These results support that the inhibition mechanism discovered in this paper might be widely used by Gram-positive bacteria.**
- 5) We have gone through the manuscript carefully to correct the inaccurate descriptions, and reorganized the results to better support our conclusions.**

The conformational changes that are reported, are certainly complex and it is unlikely that these could be predicted. However, there is at least one other TA system where (other but) equally large

changes in the toxin are observed upon binding the antitoxin, and also in that case an alternative mode of dimer formation of the toxin is observed: *E. coli* rnlAB (Garcia-Rodriguez et al, 2021, NAR 49(12):7164-7178). This work is not cited nor discussed as it should be.

Response: We thank the reviewer for pointing this out. We have cited this paper in the revised manuscript as “A recent study on RnlA-RnlB, a type II toxin-antitoxin system encoded by *E. coli* K-12 chromosome, also revealed complex conformational changes on the toxin RnlA. The RnlA dimer can shift its domain orientation and dimerization interface to make it in equilibrium between an inactive resting state and an active state. The antitoxin RnlB only recognizes the active RnlA dimer and inhibits its activity by directly binding to the catalytic cleft of the RnlA dimer, a strategy used commonly in the Type II toxin-antitoxin systems”.

Some linguistic and other issues:

line 14: "once" -> "originally"

Response: We have revised the manuscript as suggested.

line 32: "complicated" -> "complicate" (in the past tense it means the problem is now already solved).

Response: We have revised the manuscript to remove this error.

line 60: "EsaD was reported to be positioned in the membrane". It is not clear if the membrane of *S. aureus* is meant, or the membrane of a target cell

Response: We have removed this sentence from the revised manuscript.

line 67: "with the C-terminal nuclease domain and neutralize the toxicity of EsaD" -> with the C-terminal nuclease domain of EsaD and neutralize its toxicity.

Response: We have revised the manuscript as suggested.

lines 80-81: "However, secondary structure prediction indicated neither intrinsic disordered region nor DNA binding domain in EsaG"

While the crystal structures confirm these statements, it should be noted that this cannot be derived from prediction. For protein segments that fold upon binding, both secondary structure

predictors based on sequence as well as AlphaFold2 tends to predict the folded state. Also disorder predictors do not recognize folding-upon)binding IDP domains of antitoxins as unfolded.

Response: We have removed this sentence from the revised manuscript.

lines 94-95: "we launched a systematic effort to solve the structures ..." > "we determined the structures ...". Solving three structures is not a systematic effort.

Response: We have revised this sentence as “we next solved the crystal structures of EsaD and the EsaD-EsaG complex”.

line 95: "of the nuclease domain" -> "of the nuclease domain of EsaD"

Response: We have revised the manuscript as suggested.

line 111: "The conformation of EsaDc is out of expectation in the EsaDc-EsaG complex structure". This is incorrect English. Furthermore, one should not talk about a "complex structure" (which means that the structure is very complicated" but about "the structure of the complex". Please correct this throughout the text.

Response: We have revised the manuscript as suggested.

line 112: "asymmetric unit cell" -> "asymmetric unit" (an "asymmetric unit cell does not exist", you talk either about the asymmetric unit or about the unit cell, which are two distinct things that should not be mixed)

Response: We have revised the manuscript as suggested.

line 119: "the central 6-stranded beta sheet of EsaDc is split in two separate subunits". It is NOT split in SUBUNITS (that is a wrong term here). You should just write that "it is split".

Response: We thank the reviewer for pointing this out. We used NF (N-terminal fragment of EsaDc) and CF (C-terminal fragment of EsaDc) in the revised manuscript to replace Subunit1 and Subunit2, respectively.

line 121: "from the right side of $\beta 4$ ". There is no such thing as a right or a left side. Nor a top and bottom or back and front. This depends on the orientation of your molecule, and there is no universal reference frame. Therefore, this type of description is meaningless.

Response: We have revised the manuscript as suggested.

line 136: "All the secondary structures" -> "All the secondary structure elements"

Response: We have revised the manuscript as suggested.

line 163: you either "performed A His-tag pull down assay" or you "performed His-tag pull down assayS"

Response: We have revised the manuscript as “performed a His-tag pull down assay”.

Same line 170: "We also performed A cross-linking assay" or "We also performed cross-linking assayS" depending on whether there was one or more than one different cross-linking assay

Response: We have revised the manuscript as suggested.

Line 168: I do not understand how a pull-down assay can tell you anything about the stoichiometry and conformational changes that occur upon interaction between EsaD and EsaG. The cross-linking mass spec experiments are more revealing in this respect, but to be really convincing, SAXS analysis of free proteins and the complex need to be performed as well.

Response: We are sorry for the misleading language in this part. The cross-linking assay and pull-down assay were designed to validate that EsaDc interact with EsaG through two separated fragments, as observed in the EsaDc-EsaG heterodimer. We have reorganized sentences in this part to make it clear.

Lines 184-186: "Another discovery in the pull-down assay is the association between the NT and the CT (Fig 2d), we thus deduced that these two fragments may reassemble into an intact structure of EsaDc in the absence of EsaG"

This is not really surprising as it is the case for many proteins to produce a correctly folded structure out of two fragments. For example, a number of applications using GFP in cell biology are based on this property. Spending a whole paragraph on this is therefore not warranted.

Response: We agree with the reviewer that many proteins have been found to produce a correctly folded structure out of two fragments. Following the reviewer’s suggestion, we have shortened this part by removing the result of gel-shift experiment.

But we believe this is an important part to the inhibitory mechanism proposed in our paper. During secretion process, EsaG is removed from the complex. However, we don’t know whether the two separated fragments reassemble into a functional protein

spontaneously or with the help of other proteins. The results of our experiments, including pull-down assay, NMR experiment and enzymatic activity assay, lend strong support to the spontaneous assemble of EsaDc fragments.

lines 205-206: What do you mean with "is well preserved in the crystal lattice"? Do you mean the asymmetric unit contains a monomer and the dimer is formed according to crystal symmetry? If so, rephrase correctly. If you mean something else, also rephrase so the reader knows what you mean. In any case, confirmation of the dimer arrangement via SAXS is required.

Response: 1) We removed this sentence from the revised manuscript. 2) As mentioned above, we tried to follow the review's suggestion to perform SAXS at Shanghai Synchrotron Radiation Facility, but our appointment was postponed indefinitely because of the outbreak of COVID-19. It is still shut down now. We therefore performed chemical cross-linking and density gradient centrifugation experiments, which supported a monomer form of EsaG in solution. We thank the reviewer sincerely for pointing this out, which help us correct the error in our paper.

line 220: "Each of the EsaG structure in the dimer ..." This reads like in the dimer, the two EsaG monomers adopt different conformations. I think you mean "Each of the EsaG monomers in the dimer ..."

Response: We have removed this sentence from the revised manuscript.

lines 223-224: "However, severe steric clash occurs between Subunit1 and the unoverlapped EsaG in the homodimer (Fig S9)"

I do not understand this sentence. Wat is "Subunit1"? It reads like in your crystal structure the two EsaG monomers that make up the EsaG dimers clash with each other, which is a contradiction.

Response: "Subuint1" refers to the N-terminal fragment of EsaDc. Since we confirmed the monomer state of EsaG, the sentence mentioned above has been removed from the new manuscript.

Reviewer #2 (Remarks to the Author):

This manuscript presents a structure function study of a toxin antitoxin complex from the staphylococcal T7SSb. The authors provide an in-depth understanding of the molecular mechanism, by which the antitoxin EsaG inhibits the nuclease toxin EsaD. It is a very nice study which makes an important contribution to the T7SS field.

Response: We appreciate the positive comments of this reviewer on our manuscript. Our responses to the reviewer's advice on the minor points/questions are listed below.

Comments:

-the authors should briefly explain why they used a truncated version of EsaD for their crystallization studies and subsequently for all functional assays. Does the proposed inhibition mechanism also work with full-length EsaD?

Response: We thank the reviewer for pointing this out. EsaD contains two domains: the N-terminal domain of EsaD interacts with a chaperon protein, EsaE, and they together play critical roles in the secretion of EsaD through T7SS; the C-terminal domain of EsaD encodes a typical $\beta\beta\alpha$ -metal nuclease, which interacts with the antitoxin EsaG. It has been shown that the interaction of EsaG with the nuclease domain of EsaD is sufficient to neutralize the toxicity of (*Nat. Microbiol.*, 2016, 2, 16183). We also clarified this in the revised manuscript.

- In the prey cell EsaG is supposed to neutralize EsaD. How fast are EsaG-EsaD complexes formed and what is the association rate?

Response: We tested the binding affinity between EsaD and EsaG by BLI. The K_d is about 0.46 μ M. In fact, EsaD is highly toxic to the host cells. The wild-type EsaD gene is even unable to be cloned in *E. Coli.*, so the protein has to be neutralized immediately upon synthesis.

- in the cytoplasm the EsaD-EsaG complex is bound by EsaE. Could this have an effect on complex formation/inhibition?

Response: It has been reported that EsaE does not interact with EsaG or the nuclease domain of EsaD (*Nat. Microbiol.*, 2016, 2, 16183).

-The proposed heterotetramer formation is interesting but does not contribute to the inhibition mechanism itself. Could the authors speculate on its potential biological relevance in the discussion?

Response: We tried to figure out the biological functions of the heterotetramer formation by mutagenesis, but failed to obtain any mutant that disrupt the dimeric interface of EsaD while at the same time keep the protein stable. We speculate that the heterotetrameric arrangement may help keep EsaDc in the inhibited state by stabilizing the exposed hydrophobic residues on CF. We have added this in the Discussion as suggested.

- Pull down assay: it is unclear to me what was loaded onto the NiNTA column: purified proteins or cell lysates containing the overexpressed proteins. The figure legend for 2d) could use a description.

Response: Thank you pointing this out. Purified proteins were used to perform the pull-down assays. We have added this information in the corresponding figure legend.

- the Ramachandran Plot statistics should also be provided in table 1.

Response: The Ramachandran Plot statistics have been provided in Table S1 as suggested.

-the labeling is a bit inconsistent and could also be clearer. Subunit 1/2 would be better labeled as EsaDc-Subunit1/2 or perhaps better EsaDc subdomain 1/2. Pull-down assay: Perhaps use His-EsaDc-Subdomain 1/2 instead of His-Sumo-Nt/Ct as label.

Response: We are sorry about this problem. In the revised manuscript, we replaced Subunit1/2 with NF/CF, because another reviewer disagree that EsaD is split into “subunits”. As suggested, we labeled samples in the pull-down assay as His-EsaDc-NF/CF.

-the sentence in lines 79-82 should be reformulated. This sounds like as if secondary structure predictions can be used to predict DNA binding sites.

Response: We have removed this sentence from the revised manuscript.

-spelling errors: line 71, therapeutic; line 174: Å; line 286: subunit1.

Response: We have revised these errors.

Reviewer #3 (Remarks to the Author):

In the presented paper, the authors report the structure of EsaD separate, and in complex with EsaG. It is shown that significant structural rearrangements of the EsaD's c-terminal occur upon binding to EsaG. Although the results on the interactions between EsaD and EsaG are interesting, I don't think they are significant enough for Nature communications. Therefore, the paper would be a better fit in a more specialized journal. Finally, I have some comments below that should be addressed before publication.

Response: We are grateful to the reviewer for his/her advices on our manuscript. However, we want to stress that our finding is not just interesting, but also represents a major conceptual advance on the toxin-antitoxin systems.

Firstly, toxin-antitoxin (TA) modules play a central role in regulation of bacterial growth and host-pathogen interaction. The functional diversity and ubiquitous distribution of type II TA modules are causing significant attention. For most type II TA modules, inhibition of the toxin by antitoxin relies on an intrinsically disordered region of the antitoxin that refolds upon toxin binding. The structures determined in this study present a totally different strategy, in which significant conformational changes happen on the toxin, but not the antitoxin. Additionally, in contrast to the extensive studies on the TA modules from Gram-negative bacteria, few examples have been unraveled in Gram-positive bacteria. Our studies show that this unique inhibition mechanism exists commonly in the Gram-positive bacteria, which will greatly promote our understanding to the toxin-antitoxin systems.

Secondly, the conformational changes during the EsaDc-EsaG interaction is complicate, and is "unlikely that these could be predicted", as pointed out by reviewer 1. The structures determined in our paper provide new examples to study the plasticity and dynamic of proteins, which is not only attractive to researchers studying the structure-function of macromolecules, but also presents a challenge to theoretical experts who try to predict protein-protein interactions. In the revised manuscript, we performed ratchet&pawl potential and REST2 to get insights into the mechanism that is difficult to understand solely from experiments. But our work only revealed what happens at the very early stage of EsaD-EsaG association, how the β -sheet in the core of EsaDc is split remains elusive. We are still looking for top professionals on MD to analyze the process.

Thirdly, multidrug resistance among Gram-positive bacteria, especially methicillin-resistant *S. aureus* (MRSA), has been a major healthcare concern worldwide. Toxin-antitoxin modules usually play critical roles in persister formation. The inhibitory mechanism revealed in this study therefore sheds light on development of novel antibiotics that release EsaD from the complex to kill the bacteria. Furthermore, EsaD is the first antibacterial toxin identified to secrete through the T7SS of *S. aureus*. The structure of EsaDc-EsaG complex will help to design EsaD mutants that are insensitive to EsaG, which may also be used to treat or prevent *S. aureus* infection. As pointed out by reviewer 2, this would make “an important contribution to the T7SS field”.

Together, we believe that our finding is scientifically significant and will be of broad interest in readers of *Nature Communications*. The newly added data and revisions have greatly strengthened our manuscript. We hope that the reviewer could reconsider our manuscript.

1. The authors only crystallize the c-terminal of EsaD with and without EsaG. They should motivate how we can be sure that these structures are representative for the full EsaD-EsaG complex.

Response: We understand the reviewer’s concern and thank the reviewer for pointing this out. EsaD contains two domains: the N-terminal domain of EsaD interacts with a chaperon protein, EsaE, and they together play critical roles in the secretion of EsaD through T7SS; the C-terminal domain of EsaD encodes a typical $\beta\beta\alpha$ -metal nuclease, which interacts with the antitoxin EsaG. It has been shown that the interaction of EsaG with the nuclease domain of EsaD is sufficient to neutralize the toxicity of (*Nat. Microbiol.*, 2016, 2, 16183). We also clarified this in the revised manuscript.

2. The statements about the polymerization states of EsaG and EsaD through the paper can be a bit confusing. In the introduction the authors say that in the free state EsaD is a monomer and EsaG is a dimer. But in the bound state, EsaD is a dimer and EsaG is a monomer, therefore the complex should be a heterotrimer right? This is not consistent with the results paper of the section. For example in “Dimer to monomer conversion of of EsaG during EsaDc-EsaG interaction” the authors say that the EsaDc-EsaG is a heterotetramer according to gel-filtration experiments and that two EsaDc-EsaG complexes might interact. I believe the authors suggest an EsaG-EsaD-EsaD-EsaG arrangement, but then it should have been made clear in the introduction.

Response: We are sorry about this problem. Our judgments of the oligomeric state of EsaDc and EsaDc-EsaG were based on crystal structures and gel-filtration experiments. In the revised manuscript, we performed extensive studies, including native mass spectrum, density gradient centrifugation, chemical cross-linking and ¹⁹F-NMR to explore the truth. These new data identified that EsaG exists as monomer dominantly in solution, while the oligomeric state of the EsaDc-EsaG complex exists in solution and is concentration dependent. We have corrected the model in the revised manuscript accordingly. But the key discovery of this paper, the unique toxin inhibition mechanism is further solidified.

Comments on MD simulation:

Overall response: The manuscript has been revised extensively, and the umbrella sampling calculation in the previous manuscript has been replaced by the results of cross-linking experiment and ¹⁹F-NMR. Nevertheless, we thank you sincerely and followed the suggestions in the newly added ratch&pawl MD simulations and REST2 simulations.

3. The error bars for the energy profiles from Umbrella sampling look really large (Figure 4C). I suspect this is because the simulation length for each window are really short, only 5 ns. They should probably be around 10-15 ns to improve the errors.

Response: We carried out cross-linking experiment and ¹⁹F-NMR measurement to prove the EsaDc-EsaG dimers form a tetramer through their CFs, so the umbrella sampling energy profile is not included in the revised manuscript. However, we followed the suggestions in the ratch&pawl simulations and REST2 simulations. The simulation length for ratch&pawl simulations is 500 ns. For the REST2 simulations, the total sampling time is 8 μ s.

4. The authors state that bootstrap analysis was used to calculate the errors, but they need to give more details. There are several approaches to do this for umbrella sampling, and some can underestimate the errors. See for example this publication <https://pubs.acs.org/doi/pdf/10.1021/ct100494z>.

Response: Thank you for the information. We will follow it in our future studies. As mentioned above, the umbrella sampling energy profile is not included in the revised manuscript.

5. The authors say that equilibration simulations were done at the temperature of 277 K and don't say anything about the temperature in the umbrella sampling simulations. What was the temperature in these simulations? If the temperature in the umbrella sampling simulations was also 277 K the authors should motivate this choice, considering the structural changes are expected to occur at room temperature in vivo.

Response: Thank you for point this problem out. In the new simulations for the binding process of EsaG to EsaDc, the temperature was set to 300 K. In the REST2 simulations, the neutral replica for analysis was also at 300 K.

6. The authors should report the force constant that was used for window restraints in the umbrella sampling simulations.

Response: The umbrella sampling energy profile has been removed in the revised manuscript.

7. Finally, the CHARMM36 protein force field have been upgraded to CHARMM36m in 2017 and should be used in future simulations.

Response: Thanks for the comment. We used CHARMM36m for the newly added ratchet&pawl MD and REST2 simulations.

REVIEWERS' COMMENTS

Reviewer #1 (Remarks to the Author):

The authors have significantly altered their manuscript and taken care of the issues that were raised. The paper can be accepted in its current form.

Reviewer #3 (Remarks to the Author):

The authors present a revised paper on EsaD toxin and EsaG antitoxin interaction. The clarity of the paper has greatly improved in the new version. The implications of the

results are explained better, and additional experiments with YezG and YeeF to demonstrate a trend are appreciated. The authors have performed a lot of experiments and new simulations to consistently demonstrate the structural changes in EsaD upon contact with EsaG. The new simulations fit better with the rest of the paper, and my concerns in the previous version have been addressed.

In the present version the paper is a good fit for Nature Communications. However there are still minor points that should be addressed before publication.

1. I have a question about the new REST2 simulations. In the methods the authors say that “effective temperature spaced exponentially between 300 K and 450 K for residues from S444 to G558.” Do the authors mean that the temperature tempering was applied only to these residues of EsaDc, this should be explicitly stated in paper. Also according to Figure 1 EsaD has 614 residues, what was the reasoning for excluding the end of the C-terminal?

2. In view of the new experiments the authors now clarify that the EsaD-EsaG forms dimers at lower concentrations and tetramers at higher concentrations (50 uM). My question is are concentration of 50 uM biologically relevant? Please discuss this in the paper.

Minor comments

3. Line 14 abstract, "This is totally different from that of...". "This" is unspecific by itself and "totally" is a bit too informal for the abstract. I propose to change it to "This mechanism is distinct from that of...".

4. Line 652 methods, the force constant unit should be kcal/mol/A² based on the formula for the restraining potential.

5. Figure captions in general. Coloring is not explained in Figures after Figure 1. The authors should either explain colors in all figures or refer to Figure 1 one in the caption to improve readability.

6. Figure 3A caption, I would mention the magnesium ion and its representation here for clarity.

We thank both the reviewers for their comments on our work. Please find our point-to-point response to each of the reviewers' comments below.

REVIEWER COMMENTS

Reviewer #1 (Remarks to the Author):

The authors have significantly altered their manuscript and taken care of the issues that were raised. The paper can be accepted in its current form.

Response: We are immensely grateful to the reviewer for supporting the publication of our paper.

Reviewer #3 (Remarks to the Author):

In the present version the paper is a good fit for Nature Communications. However there are still minor points that should be addressed before publication.

Response: We are grateful to the reviewer for his/her advices on our manuscript and for supporting the publication of our paper. We have revised the manuscript carefully following the reviewer's suggestions.

1. I have a question about the new REST2 simulations. In the methods the authors say that "effective temperature spaced exponentially between 300 K and 450 K for residues from S444 to G558." Do the authors mean that the temperature tempering was applied only to these residues of EsaDc, this should be explicitly stated in paper. Also according to Figure 1 EsaD has 614 residues, what was the reasoning for excluding the end of the C-terminal?

Response: As proposed in the paper, EsaG interacts with CF (residue 559-614) of EsaDc first to trigger conformational changes in the NF and the $\beta\beta\alpha$ -metal finger motif (residue 444-558). During the interaction, CF undergoes no obvious conformational change. To maximize the efficiency of REST2 sampling, we only included the N-terminal region in the simulation. We also made it clear in the Methods.

2. In view of the new experiments the authors now clarify that the EsaD-EsaG forms dimers at lower concentrations and tetramers at higher concentrations (50 μ M). My question is are concentration of 50 μ M biologically relevant? Please discuss this in the paper.

Response: We understand the reviewer's concern. Following the reviewer's suggestion, we discussed the biological relevance of EsaD-EsaG tetramers in the Discussion as "For example, what is the biological relevance of the EsaD-EsaG heterotetramer? The concentration of EsaD-EsaG complex has never been measured in the living bacteria, making it hard to assess the impact of the EsaD-EsaG heterotetramer. Since EsaD is secreted into surrounding environment to kill competitors, it is highly possible that high level of EsaD is expressed and the heterotetrameric arrangement of EsaD-EsaG may help keep EsaD in the inhibited state before secretion."

3. Line 14 abstract, "This is totally different from that of...". "This" is unspecific by itself and "totally" is a bit too informal for the abstract. I propose to change it to "This mechanism is distinct from that of...".

Response: We have revised this sentence as suggested.

3. Line 652 methods, the force constant unit should be kcal/mol/A² based on the formula for the restraining potential.

Response: Thank you for pointing this error out. It has been revised in the new manuscript.

4. Figure captions in general. Coloring is not explained in Figures after Figure1. The authors should either explain colors in all figures or refer to Figure 1 one in the caption to improve readability.

Response: Thank you for the suggestion. We have explained the colors in all the figures.

5. Figure 3A caption, I would mention the magnesium ion and its representation here for clarity.

Response: We have added the information in the figure legend.